# Alleviating Sparse Rewards by Modeling Step-Wise and Long-Term Sampling Effects in Flow-Based GRPO

Yunze Tong [1] [*]   Mushui Liu [1] [2] [*] [†]   Canyu Zhao [1]   Wanggui He [2]   Shiyi Zhang [3]   Hongwei Zhang [1]   Peng Zhang [2]   Jinlong Liu [2]   Hao Jiang [2] [†]

## Abstract

Deploying GRPO on Flow Matching models has proven effective for text-to-image generation. However, existing paradigms typically propagate an outcome-based reward to all preceding denoising steps without distinguishing the local effect of each step. Moreover, current group-wise ranking mainly compares trajectories at matched timesteps and ignores within-trajectory dependencies, where certain early denoising actions can affect later states via delayed, implicit interactions. We propose TurningPoint-GRPO (TP-GRPO), a GRPO framework that alleviates step-wise reward sparsity and explicitly models long-term effects within the denoising trajectory. TP-GRPO makes two key innovations: (i) it replaces outcome-based rewards with step-level incremental rewards, providing a dense, step-aware learning signal that better isolates each denoising action's "pure" effect, and (ii) it identifies turning points—steps that flip the local reward trend and make subsequent reward evolution consistent with the overall trajectory trend—and assigns these actions an aggregated long-term reward to capture their delayed impact. Turning points are detected solely via sign changes in incremental rewards, making TP-GRPO efficient and hyperparameter-free. Extensive experiments also demonstrate that TP-GRPO exploits reward signals more effectively and consistently improves generation. Code is available at https://github.com/YunzeTong/TurningPoint-GRPO.

[*]Equal contribution  [1]Zhejiang University, Hangzhou, China [2]Taobao & Tmall Group of Alibaba, Hangzhou, China [3]Tsinghua University, Beijing, China. [†]Correspondence to: Mushui Liu, Hao Jiang <lms@zju.edu.cn, aoshu.jh@taobao.com>.

*Proceedings of the 43$^{rd}$ International Conference on Machine Learning*, Seoul, South Korea. PMLR 306, 2026. Copyright 2026 by the author(s).

## 1. Introduction

Flow Matching (FM) models (Lipman et al., 2022; Liu et al., 2022) can transport a simple prior to a complex target distribution via the learned velocity field, and have been widely adopted for text-to-image generation. Motivated by recent progress in large language models, researchers have applied Group Relative Policy Optimization (GRPO) (Shao et al., 2024; Guo et al., 2025) to FM models. Representative methods such as Flow-GRPO (Liu et al., 2025a) and DanceGRPO (Xue et al., 2025) evaluate a reward on the final generated image and assign this same terminal reward to each preceding denoising step produced by a Stochastic Differential Equation (SDE)-based sampler (Song et al., 2021). Advantages computed from these per-step rewards enable effective post-RL fine-tuning and can improve overall performance.

However, this design does not accurately model step-level reward assignment. It leads to two issues: (1) The reward reflects the aggregate effect of the entire denoising trajectory and is identically assigned to every step, without distinguishing the contribution of different steps, which induces reward sparsity. (2) Existing methods mainly leverage trajectory-level ranking across sampled trajectories, while neglecting within-trajectory interactions among steps, even though such dependencies are crucial for composing a coherent sample.

We illustrate the first issue by sampling several trajectories and visualizing their step-wise reward evolution in Figure 1. For a latent at the intermediate step, we perform Ordinary Differential Equation (ODE) sampling for the remaining steps to obtain a clean image on which the reward can be evaluated. This procedure is motivated by the fact that, compared to SDE sampling, ODE sampling removes stochasticity while preserving the same marginal distributions (Song et al., 2021); thus, the ODE completion can be interpreted as an average over possible SDE outcomes from the same intermediate state. With this estimator, Figure 1 shows that the reward can oscillate frequently during denoising. In contrast, Flow-GRPO assigns only the reward of clean images to all preceding steps, which captures the relative ordering of complete trajectories but may conflict with local progress. For example, from $t = 6$ to $t = 5$, the orange and green trajectories exhibit a local reward decrease; yet because their

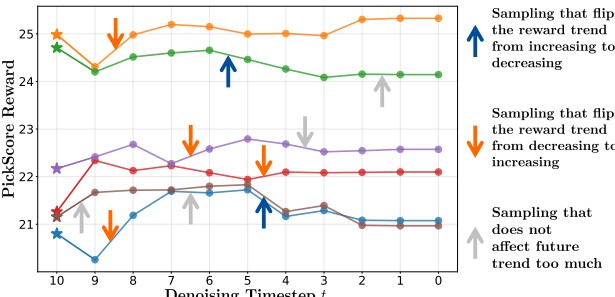

*Figure 1.* Rewards of several sampled trajectories. Each dot at $t$ is obtained by $(10 - t)$ steps of SDE sampling followed by $t$ steps of ODE sampling. The leftmost point corresponds to full ODE sampling, and the rightmost to full SDE sampling (*i.e.*, standard Flow-GRPO outputs).

full SDE-based samples achieve higher terminal rewards at $t = 0$, they receive larger advantages for this step, incorrectly reinforcing an action that locally degrades reward. Overall, this outcome-based reward allocation cannot isolate the "pure" gain of each denoising step. It therefore induces step-level reward sparsity and global–local misalignment, which can limit the effectiveness of RL fine-tuning.

The second issue stems from the sequential nature of FM generation. Each denoising action affects not only the immediate next latent, but also subsequent denoising behavior and thus the future trajectory. An intermediate state $x_t$ depends implicitly on earlier states $x_{t+2}, \dots, x_T$, since different upstream trajectories may lead to different $x_{t+1}$ and thereby change the starting point of the current update. We refer to this delayed dependence as *implicit interaction*. This effect is also reflected in the reward dynamics: in Figure 1, the local reward trend can temporarily deviate from the overall trend from noise to the final image, and later be restored by a critical step that flips the trend. We call such steps *turning points*, which are common in practice (*e.g.*, $t = 9$ on the blue trajectory, $t = 6$ on the green trajectory, and $t = 7$ on the purple trajectory). Their impact is not limited to the immediate step reward, but also shapes later reward evolution until the end of denoising. However, GRPO forms groups by aligning steps at the same $t$, which mainly supports cross-trajectory ranking but ignores within-trajectory dynamics induced by implicit interaction, leaving the rewards of such key turning points under-modeled.

To address these issues, we propose **T**urning**P**oint-GRPO (TP-GRPO), a framework that aligns the nature of FM and enhances Flow-based GRPO. TP-GRPO has two folds. First, to reduce the misalignment between local steps and the overall trajectory, we introduce an **incremental-effect-based step-wise reward** that replaces the sparse outcome-based reward. Concretely, our step-aware reward is defined as the difference between rewards obtained after and before a single SDE sampling, thereby more faithfully reflecting the relative quality of individual denoising actions within each group. Second, to explicitly account for the delayed ef-

fects of key steps—an aspect not modeled in prior work-we further design an **aggregation-based implicit interaction modeling** mechanism. We formally define *turning points* as steps that flip the local reward trend, such that taking SDE sampling at that step makes the subsequent reward evolution consistent with the overall trend. For these turning-point actions, we assign an aggregated long-term reward as feedback. This encourages directions that are likely to improve future reward trends and discourages those leading to overall reward drops. Crucially, turning points are identified solely by sign changes in incremental rewards, not their magnitudes, making our method efficient and hyperparameter-free.

We summarize our contributions as follows:

- We identify reward sparsity and step-level misalignment caused by propagating an outcome-based reward to intermediate denoising steps. We address this by using step-wise reward differences to capture the incremental effect of each SDE update, yielding a better estimate of a single step's "pure" gain.

- Based on this fine-grained signal, we uncover turning points—steps that flip the local reward trend to match the overall trajectory. We provide a strict sign-based criterion to identify them and assign aggregated long-term rewards to model their delayed impact. To our knowledge, this is the first work to explicitly model such implicit interaction in Flow-based GRPO.

- Building on these insights, we propose **TurningPoint-GRPO**, which mitigates reward sparsity and improves delayed credit assignment. Extensive experiments further demonstrate the effectiveness of our method.

## 2. Related Work

**Diffusion-based Models.** Diffusion-based models have been widely adopted in a broad range of generative modeling tasks, including image generation (Ying et al., 2024; Liu et al., 2025b; Mushui Liu, 2025; Zhao et al., 2025a; Hu et al., 2025a; Tong et al., 2025b), video generation (Zhao et al., 2024; She et al., 2025), 3D generation (Zhao et al., 2025b;c; Tan et al., 2026), and tabular data synthesis (Zhang et al., 2024; Tong et al., 2025a). These models learn data distributions through a gradual noising and denoising process, and have demonstrated strong performance across diverse domains. Their probabilistic formulation provides a flexible framework for modeling complex and high-dimensional data, contributing to their growing adoption in recent years.

**Common RL Techniques.** Reinforcement learning (RL) has become a central tool for aligning large language models (LLMs) with human preferences and downstream objectives (Guo et al., 2025; Jaech et al., 2024). A standard pipeline first trains a reward model from human preference

data, and then optimizes the policy using proximal policy optimization (PPO) (Schulman et al., 2017). To better exploit multiple candidates per prompt, group-based ranking objectives have been proposed. In particular, GRPO (Shao et al., 2024) and its variants (Zheng et al., 2025a; Yu et al., 2025) have been widely adopted for LLM post-training.

**RL for Diffusion and Flow Models.** Motivated by these advances in LLM alignment, recent work has explored RL for diffusion-based models. Several methods apply Direct Preference Optimization (DPO) (Yang et al., 2024; Wallace et al., 2024; Hu et al., 2025c;b; Liang et al., 2025) or PPO-style algorithms (Black et al., 2023; Fan et al., 2023; Miao et al., 2024; Tong et al., 2025c; Zhang et al., 2026) to modify the denoising trajectories. More recently, some methods (Liu et al., 2025a; Zheng et al., 2025b) adapt GRPO-style objectives for fine-tuning flow models. This line of work (He et al., 2025; Li et al., 2025; Deng et al., 2026; Zhao et al., 2026) introduces stochasticity via SDE-based sampling during denoising to form candidate groups and encourage exploration, leading to improved performance.

## 3. Preliminary

### 3.1. Flow Matching

Flow Matching (Lipman et al., 2022) learns a time-dependent model $v_\theta(x_t, t)$ that approximates the velocity field transporting a prior $p_0$ to the target distribution $p_1$. Sampling is performed by integrating the deterministic ODE

$$\mathrm{d}x_t = v_\theta(x_t, t)\,\mathrm{d}t. \tag{1}$$

Recent works (Liu et al., 2025a; Xue et al., 2025) convert this ODE sampler into an equivalent SDE sampler. For rectified flow (Liu et al., 2022), the SDE-based update rule is

$$x_{t+\Delta t} = x_t + \left[ v_\theta(x_t, t) + \frac{\sigma_t^2}{2t}\big(x_t + (1-t)v_\theta(x_t, t)\big) \right] \Delta t + \sigma_t \sqrt{\Delta t}\,\epsilon, \tag{2}$$

where $\sigma_t = \alpha\sqrt{\frac{t}{1-t}}$ and $\epsilon \sim \mathcal{N}(0, I)$. $\alpha$ is a scalar hyper-parameter for noise level control. Previous works (Song et al., 2021; Karras et al., 2022) have also shown that an SDE sampler can induce the same marginal distributions at each noise level as its corresponding ODE sampler. The key difference is that stochasticity in the ODE sampler arises solely from the random initialization, whereas in the SDE sampler it comes from both the initial noise and the diffusion term along the trajectory.

### 3.2. Adopting SDE-sampler in FM for GRPO

GRPO is an online RL method that leverages the contrast among self-sampled trajectories to guide policy exploration. A key requirement for applying GRPO is to obtain diverse rollouts conditioned on the same context or input. To deploy it on flow matching, current methods (Liu et al., 2025a) adopt the SDE sampling in Eq. 2 to inject stochasticity into the denoising process, thereby producing diverse trajectories and corresponding final images. Taking Flow-GRPO as an example, given a prompt $c$, the flow model $p_\theta$ samples a group of $G$ individual images $\{x_0^i\}_{i=1}^G$ and the corresponding reverse-time trajectories $\{(x_T^i, x_{T-1}^i, \cdots, x_0^i)\}_{i=1}^G$. Then, the advantage of the $i$-th image is computed by normalizing the group-level rewards:

$$\hat{A}_t^i = \frac{R(x_0^i, c) - \mathrm{mean}(\{R(x_0^i, c)\}_{i=1}^G)}{\mathrm{std}(\{R(x_0^i, c)\}_{i=1}^G)}. \tag{3}$$

The policy model is then optimized by maximizing:

$$\mathcal{J}_{\text{Flow-GRPO}}(\theta) = \mathbb{E}_{c \sim \mathcal{C}, \{x^i\}_{i=1}^G \sim \pi_{\theta_{\text{old}}}(\cdot|c)} f(r, \hat{A}, \theta, \varepsilon, \beta). \tag{4}$$

With $r_t^i(\theta) = \frac{p_\theta(x_{t-1}^i|x_t^i, c)}{p_{\theta_{\text{old}}}(x_{t-1}^i|x_t^i, c)}$, the computation becomes:

$$f(r, \hat{A}, \theta, \varepsilon, \beta) = \frac{1}{G}\sum_{i=1}^G \frac{1}{T}\sum_{t=0}^{T-1} \Bigg( \min\Big(r_t^i(\theta)\hat{A}_t^i,$$
$$\mathrm{clip}\Big(r_t^i(\theta), 1-\varepsilon, 1+\varepsilon\Big)\hat{A}_t^i\Big) - \beta D_{\text{KL}}(\pi_\theta||\pi_{\text{ref}}) \Bigg). \tag{5}$$

## 4. Analysis on Limitations of Terminal Reward Assignment

### 4.1. Reward Sparsity and Misaligned Per-Step Reward Allocation

In this subsection, we identify the problem of reward sparsity, which leads to inaccurate stepwise credit assignment. For each trajectory in a group, the reward is computed only once from the final clean image $x_0^i$, and this single scalar is then assigned uniformly to all denoising steps along the trajectory. Consequently, once a group is fixed, the relative ordering of advantages computed in Eq. 3 is constant at each timestep. This reward design naturally induces sparsity, since Flow-GRPO optimizes all timesteps under SDE sampling while receiving supervision only from the final outcome. The terminal reward reflects cumulative feedback over the entire trajectory and is per-timestep agnostic, with no direct dependency on the individual denoising actions at each step. Moreover, using the final reward and its normalized advantage as a surrogate for all intermediate steps yields inaccurate contribution attribution. Prior works have shown that denoising steps at different timesteps contribute unequally to the final generation (Karras et al., 2024; Kynkäänniemi et al., 2024). Uniformly assigning the same reward to every timestep implicitly assumes identical contribution from each denoising update, ignoring the heterogeneity of denoising operations and consequently overestimating or underestimating the influence of certain timesteps in the overall generation process.

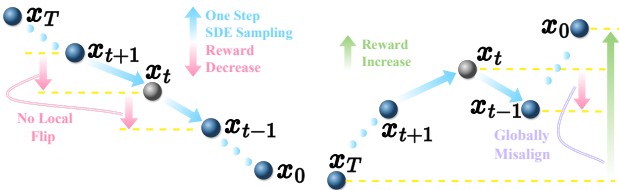

*(a)* Normal point that does not satisfy the local flip condition. *(b)* Normal point that does not align with the overall trend.

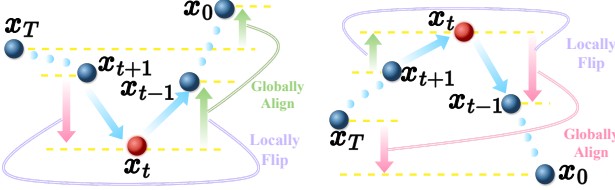

*(c)* Good turning point that changes a decreasing trend to an increasing one. *(d)* Bad turning point that changes an increasing trend to a decreasing one.

*Figure 2.* Some cases that are or are not identified as turning points. The first row shows cases that do not satisfy our turning-point definition and are optimized with $r_t$. The second row shows cases that do satisfy it and are optimized with $r_t^{\mathrm{agg}}$.

### 4.2. Turning-Point Effects and Insufficient Implicit Cross-Step Interaction Modeling

In this subsection, we show that, due to the iterative denoising nature of FM, there exists an implicit interaction across steps. This interaction reflects the aggregated effect of a subset of denoising steps and influences the final image through an indirect mechanism, which should be taken into account when assigning rewards to different timesteps in GRPO, but is not explicitly modeled in prior methods.

In flow matching, the generative denoising process in discrete time can be viewed as a Markov Decision Process-like procedure where the state is the latent representation and the action is a denoising step. Since each updated latent serves as the input to the next step, the effect of early denoising actions propagates and accumulates over time, thereby influencing subsequent updates and the final generated image.

We use Figure 1 to illustrate the implicit interaction. In Figure 1, each line represents a 10-step SDE-sampling-based denoising trajectory, and the dots at $t = 0$ indicate the rewards of the final clean images. To obtain rewards for all intermediate states, for each intermediate latent in a trajectory, we complete the remaining steps using ODE sampling to obtain a corresponding image. The reward of this image reflects the cumulative gain of all preceding SDE steps. Following Section 3.1, ODE sampling is deterministic and preserves the same marginal distribution as SDE sampling. For a latent $x_t$, we use $x_t^{\mathrm{ODE}(k)}$ to denote the latent obtained by running $k$ ODE sampling steps after the existing $(T - t)$ SDE sampling steps. When $k = t$, $x_t^{\mathrm{ODE}(k)}$ becomes a fully denoised clean image. Hence, $x_t^{\mathrm{ODE}(1)}$ can be viewed as

the statistical average of all possible SDE-sampled $\{x_{t-1}\}$, except that $x_{t-1}$ is obtained via SDE sampling from $t$ to $t - 1$, whereas $x_t^{\mathrm{ODE}(1)}$ uses ODE sampling for that step.

In Figure 1, we observe that, for most trajectories, the rewards of intermediate latents oscillate and are non-monotonic. For clarity, we use $R(x_t^{(t)})$ to denote $R(x_t^{\mathrm{ODE}(t)}, c)$. To explain the implicit interaction, we first define the consistency between a single SDE sampling step at $t$ and the overall trajectory as

$$s_t = \mathrm{sign}\big(R(x_{t-1}^{(t-1)}) - R(x_t^{(t)})\big) \cdot \mathrm{sign}\big(R(x_0^{(0)}) - R(x_T^{(T)})\big), \tag{6}$$

where $R(x_0^{(0)})$ denotes the reward of the image sampled using only SDE, and $R(x_T^{(T)})$ denotes the reward of the image obtained with only ODE. $s_t$ measures whether the local sampling action at timestep $t$ is aligned with the overall reward trend of the SDE-based trajectory. Based on this, we define the concept of a turning point as follows.

**Definition 4.1.** For an SDE-based trajectory $\{x_T, x_{T-1}, \ldots, x_1, x_0\}$ with the corresponding intermediate images $\{x_T^{\mathrm{ODE}(T)}, x_{T-1}^{\mathrm{ODE}(T-1)}, \ldots, x_1^{\mathrm{ODE}(1)}, x_0^{\mathrm{ODE}(0)}\}$, a timestep $t$ is a turning point if and only if $s_{t+1} < 0$, $s_t > 0$, $\big(\mathrm{sign}\big(R(x_{t-1}^{(t-1)}) - R(x_t^{(t)})\big) \cdot \mathrm{sign}\big(R(x_0^{(0)}) - R(x_t^{(t)})\big)\big) > 0$ and $1 \leq t \leq T - 1$.

Intuitively, a turning point is the step at which the local reward trend flips so as to become consistent with the overall trend. Figures 2a and 2b show examples that are not treated as turning points, as they violate the conditions of local trend reversal and alignment with the global trend, respectively.

Taking Figure 2c as an example, the overall trend of this trajectory improves as more SDE steps are applied. However, the sampling from $x_{t+1}$ to $x_t$ decrease the reward. The actual change in trend is induced by the sampling at $t$: the step just before it causes degradation, while this step is the one that reverses the trend and makes it consistent with the overall trajectory. Afterward, most subsequent steps improve the reward rather than harming it. In this sense, $t$ corresponds to an implicit interaction that affects subsequent sampling and contributes to the final reward. The benefit of this step is not only immediate but also propagates forward and accumulates through later steps influenced by it.

In GRPO, such twisting actions that flip the trend deserve more positive (or negative) rewards to reflect their implicit long-term impact, thereby receiving stronger preference (or rejection) during the group normalization in Eq. 3. However, to the best of our knowledge, existing methods do not explicitly model this type of implicit interaction. This motivates us to explicitly integrate interaction-aware information into the reward assignment to achieve better performance.

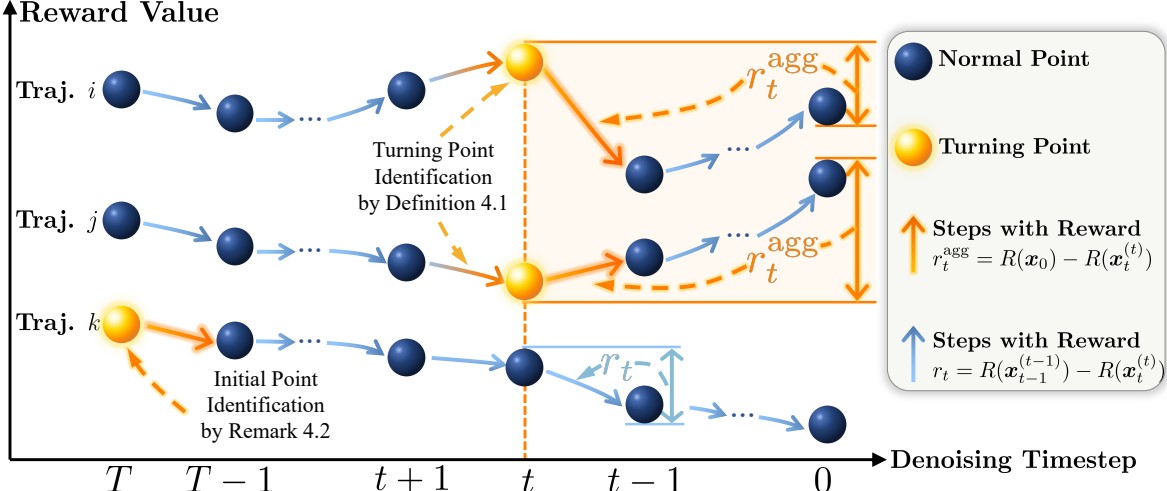

*Figure 3.* Overview of our method. For each trajectory, we compute stepwise rewards as the pure incremental effect of the current SDE sampling. We then identify the orange turning point that satisfies Definition 4.1 or Remark 5.2. Next, we assign cumulative rewards to capture their implicit impact on reversing the reward trend. Finally, we apply group normalization independently at each timestep.

## 5. Methodology

### 5.1. Increment-Based Step-wise Reward Computation

Section 4.1 has introduced the reward sparsity problem in standard GRPO-based FM. The core idea of this family of approaches is to aggregate the contribution of all sampling steps and define the reward solely in terms of the resulting global effect. However, this design is fundamentally misaligned with the need for step-wise feedback, and thus inherently induces reward sparsity. The relative ordering of advantages based on the final image reward is deterministic and may be inconsistent with the true ordering of certain local steps. Motivated by recent works (Deng et al., 2026; He et al., 2025), we instead consider replacing this global effect with the incremental change at each step and using this increment as the step-wise reward.

Building on the properties in Section 4.2, we represent $r_t$, the reward of the sampling action from $t$ to $t-1$, as the difference in gain before and after this step. Specifically, we cache the intermediate latents $\boldsymbol{x}_t$ and $\boldsymbol{x}_{t-1}$, then apply ODE sampling for $t$ and $t-1$ steps respectively to obtain $\boldsymbol{x}_t^{\mathrm{ODE}(t)}$ and $\boldsymbol{x}_{t-1}^{\mathrm{ODE}(t-1)}$. Both latents undergo a full $T$-step process, so the reward model can directly evaluate them, and they differ only in the sampling operation at timestep $t$. We then compute the effective increment

$$r_t = R\big(\boldsymbol{x}_{t-1}^{\mathrm{ODE}(t-1)}\big) - R\big(\boldsymbol{x}_t^{\mathrm{ODE}(t)}\big). \qquad (7)$$

Note that $R\big(\boldsymbol{x}_{t-1}^{\mathrm{ODE}(t-1)}\big)$ and $R\big(\boldsymbol{x}_t^{\mathrm{ODE}(t)}\big)$ share the same $(T-t)$ SDE sampling; they begin to differ at the $t$-th step. As discussed in Section 3.1, compared to SDE sampling, ODE sampling preserves the same marginal distribution and only removes stochasticity. Thus, the ODE-sampled outcome can be viewed as a statistical average and serves

as a good proxy for the baseline gain of appending the sampling at this step. By replacing the rewards in Eq. 3 with $r_t$, we can obtain more accurate step-wise rewards for intermediate sampling steps.

### 5.2. Aggregation-Based Implicit Interaction Modeling

In Section 4.2, we identify the problem of reward oscillation when deploying SDE sampling at different timesteps. We also show that turning points exert an implicit influence on subsequent sampling, which is not modeled by existing methods. We address this gap with the following design. Specifically, for timesteps that satisfy the definition of turning points, we replace the local stepwise reward with an aggregated multi-step reward, making the final advantage computation implicit-interaction-aware. For an SDE-based trajectory $\{\boldsymbol{x}_T, \boldsymbol{x}_{T-1}, \ldots, \boldsymbol{x}_1, \boldsymbol{x}_0\}$ with the corresponding images $\{\boldsymbol{x}_T^{\mathrm{ODE}(T)}, \boldsymbol{x}_{T-1}^{\mathrm{ODE}(T-1)}, \ldots, \boldsymbol{x}_1^{\mathrm{ODE}(1)}, \boldsymbol{x}_0\}$, the aggregated reward is defined as

$$r_t^{\mathrm{agg}} = R\big(\boldsymbol{x}_0\big) - R\big(\boldsymbol{x}_t^{\mathrm{ODE}(t)}\big), \qquad (8)$$

where $R\big(\boldsymbol{x}_0\big)$ denotes the reward of the final image sampled with all SDE. Compared to $r_t$, $r_t^{\mathrm{agg}}$ captures the cumulative effect from the turning point to the end of denoising. By replacing $r_t$ with $r_t^{\mathrm{agg}}$ at turning points, we also encode the implicit reward induced by their downstream impact. We provide two examples in Figure 3 to illustrate the benefit of using $r_t^{\mathrm{agg}}$ at turning points. (1) For step $t$ in trajectory $j$, the reward trend switches from decreasing to increasing, yet the local gain is very small, failing to reflect the importance of the denoising action at this step. In contrast, $r_t^{\mathrm{agg}}$ takes a larger value and more accurately reflects its contribution. (2) While for step $t$ in the trajectory $i$, the local reward exhibits a large drop, while the final reward remains close to the full ODE-based result. In this case, computing the advantage

with $r_t$ would overemphasize the effect of this step, whereas using $r_t^{\text{agg}}$ yields a more moderate contribution.

As explained above, for certain turning points identified in Definition 4.1, $r_t^{\text{agg}}$ may have a smaller absolute value than $r_t$. Building on this, we design an alternative strategy that retains only those turning points satisfying $\|r_t^{\text{agg}}\| > \|r_t\|$. This leads to a stricter notion of turning points.

**Definition 5.1.** For an SDE-based trajectory $\{\boldsymbol{x}_T, \boldsymbol{x}_{T-1}, \ldots, \boldsymbol{x}_1, \boldsymbol{x}_0\}$ with the corresponding interme­diate images $\{\boldsymbol{x}_T^{\text{ODE}(T)}, \boldsymbol{x}_{T-1}^{\text{ODE}(T-1)}, \ldots, \boldsymbol{x}_1^{\text{ODE}(1)}, \boldsymbol{x}_0^{\text{ODE}(0)}\}$, a timestep $t$ is a *consistent turning point* if and only if $s_{t+1} < 0$, $s_t > 0$, $\big(\text{sign}\big(R(\boldsymbol{x}_{t-1}^{(t-1)}) - R(\boldsymbol{x}_t^{(t)})\big) \cdot \text{sign}\big(R(\boldsymbol{x}_0^{(0)}) - R(\boldsymbol{x}_{t-1}^{(t-1)})\big)\big) > 0$, and $1 \leq t \leq T - 1$.

The steps selected by Definition 5.1 form a subset of those selected by Definition 4.1. We provide some lemmas and proofs in Appendix C. The benefit of Definition 5.1 is that it filters out "purer" implicit interactions, where the direction of the implicit interaction is aligned with the local update.

Since both $r_t$ and $r_t^{\text{agg}}$ are defined as differences of rewards, *i.e.*, as "incremental" quantities, they can be substituted for each other without introducing scale mismatch in the reward values. Moreover, the selected turning points may either improve or degrade the reward trend. Thus, we explicitly in­corporate both preference-inducing and rejection-inducing operations when computing group advantages, thereby pro­viding more reliable optimization signals for the RL process.

To sum up, the motivation for using $r_t^{\text{agg}}$ as the reward at turning points is to model and incorporate the implicit long-term impact of key SDE-based denoising actions. With this design, we expect the model to perform sampling while accounting for its potential future impact: it learns to prefer actions whose downstream trajectory is likely to enhance the final reward, and to avoid actions whose later effects would ultimately degrade it.

### 5.3. Implicit Long-Term Effect Modeling for the Initial Sampling Step

In this subsection, we propose a practical scheme for select­ing the initial sampling step to model its implicit long-term influence. This design is motivated by the observation that, under our definition of turning points, the first denoising step is naturally excluded from aggregation-based effect propagation. The proposed constraint closes this gap and yields a more comprehensive treatment of long-term effects.

Recall that our turning-point criterion relies on the auxiliary judgment based on the last denoising action before the cur­rent state ($s_{t+1} < 0$). Directly applying it omits the first denoising step ($t \leq T - 1$). Consequently, all initial sam­pling steps at $t = T$ in each trajectory receive only the local reward $r_t$ and cannot benefit from the aggregated reward

$r_t^{\text{agg}}$. However, some early steps can strongly influence the overall trajectory and their implicit effects, if modeled, can further benefit the RL process. To address this, we add a constraint that extends long-term effect detection to the first step. Concretely, we identify the initial step as eligible for using $r_t^{\text{agg}}$ as follows.

*Remark* 5.2. The sampling action at the first denoising step is selected if and only if $\big(\text{sign}\big(R(\boldsymbol{x}_{T-1}^{(T-1)}) - R(\boldsymbol{x}_T^{(T)})\big) \cdot \text{sign}\big(R(\boldsymbol{x}_0^{(0)}) - R(\boldsymbol{x}_T^{(T)})\big)\big) > 0$.

This condition selects early steps whose local reward change aligns with the overall trend, thereby capturing the impact of early decisions. In this case, initial action that strongly steers the trajectory can be assigned a more representative cumula­tive reward, better leveraging these indicative points.

## 6. Experiments

### 6.1. Experimental Setting

Our experimental setup follows Flow-GRPO. We train mod­els on three tasks: (1) *Compositional Image Generation* with Geneval (Ghosh et al., 2023) rewards; (2) *Human Pref­erence Alignment* with PickScore (Kirstain et al., 2023) rewards; and (3) *Visual Text Rendering* with a rule-based OCR accuracy reward (Chen et al., 2023; Gong et al., 2025). We adopt SD3.5-M (Esser et al., 2024) as the base model and apply LoRA (Hu et al., 2022) for efficient fine-tuning. Key hyperparameters are aligned with Flow-GRPO: training with sampling timesteps $T = 10$, inference with $T = 40$, group size $G = 24$, and image resolution as 512. All results reported in our tables and figures are obtained from our own re-implementation under a controlled and consistent config­uration on our hardware, rather than directly copied from the original Flow-GRPO paper, ensuring a fair comparison. More experimental details are provided in Appendix B.

### 6.2. Main Results

We present the quantitative comparison results in Table 1. The two TP-GRPO variants differ in whether they use Defi­nition 4.1 or Definition 5.1 for turning-point selection. With the step-level reward computation in Section 5.1 and turning-point detection in Section 5.2, TP-GRPO consistently out­performs Flow-GRPO across all three tasks (columns 2–4), while largely preserving generalization performance (columns 5–9). The qualitative comparison is provided in Figure 5. We observe that both variants of our proposed TP-GRPO improve generation quality by producing more accurate counts, better text rendering, and enhanced aesthet­ics and content alignment. We also find no indication of reward hacking in the generated images. For more qualita­tive results, please refer to Appendix G.

In addition to the best-evaluation results reported in the

*Table 1.* Results on Compositional Image Generation, Visual Text Rendering, and Human Preference benchmarks, evaluated by task scores on test prompts, and by image quality and preference scores on DrawBench prompts. **All results are from our own re-implementation under a consistent setup, not directly taken from the Flow-GRPO paper.** ImgRwd: ImageReward; UniRwd: UnifiedReward

| Model | Task Metric | | | Image Quality | | Preference Score | | |
|---|---|---|---|---|---|---|---|---|
| | **GenEval** | **OCR Acc.** | **PickScore** | **Aesthetic** | **DeQA** | **ImgRwd** | **PickScore** | **UniRwd** |
| SD3.5-M | 0.6029 | 0.4548 | 21.44 | 5.380 | 3.829 | 0.6039 | 21.95 | 3.200 |
| *Compositional Image Generation* | | | | | | | | |
| Flow-GRPO | 0.9673 | — | — | 5.223 | 3.861 | 0.8792 | 22.20 | 3.459 |
| TempFlow-GRPO ‡ | 0.9703 | — | — | 5.075 | 3.403 | 0.8574 | 21.51 | 3.201 |
| TP-GRPO (w/o constraint) | 0.9714 | — | — | 5.352 | 3.963 | 0.9267 | 22.28 | 3.491 |
| TP-GRPO (w constraint) | 0.9725 | — | — | 5.246 | 3.977 | 0.8710 | 22.20 | 3.488 |
| *Visual Text Rendering* | | | | | | | | |
| Flow-GRPO | — | 0.9579 | — | 5.091 | 3.459 | 0.6906 | 21.91 | 3.088 |
| TempFlow-GRPO ‡ | — | 0.9693 | — | 5.086 | 2.762 | 0.6316 | 21.35 | 3.017 |
| TP-GRPO (w/o constraint) | — | 0.9718 | — | 5.092 | 3.251 | 0.6433 | 21.93 | 3.160 |
| TP-GRPO (w constraint) | — | 0.9651 | — | 5.111 | 3.480 | 0.7378 | 22.06 | 3.260 |
| *Human Preference Alignment* | | | | | | | | |
| Flow-GRPO | — | — | 24.02 | 6.231 | 3.966 | 1.3875 | 24.10 | 3.605 |
| TempFlow-GRPO ‡ | — | — | 24.45 | 6.300 | 3.855 | 1.3799 | 24.25 | 3.532 |
| TP-GRPO (w/o constraint) | — | — | 24.73 | 6.293 | 3.961 | 1.3714 | 24.46 | 3.600 |
| TP-GRPO (w constraint) | — | — | 24.67 | 6.321 | 3.993 | 1.4419 | 24.61 | 3.640 |

‡ We restrict TempFlow-GRPO to its branching mechanism, excluding orthogonal enhancements to isolate the effect of process reward. The reported results are based on our implementation under its default settings, aligned with our experimental setup.

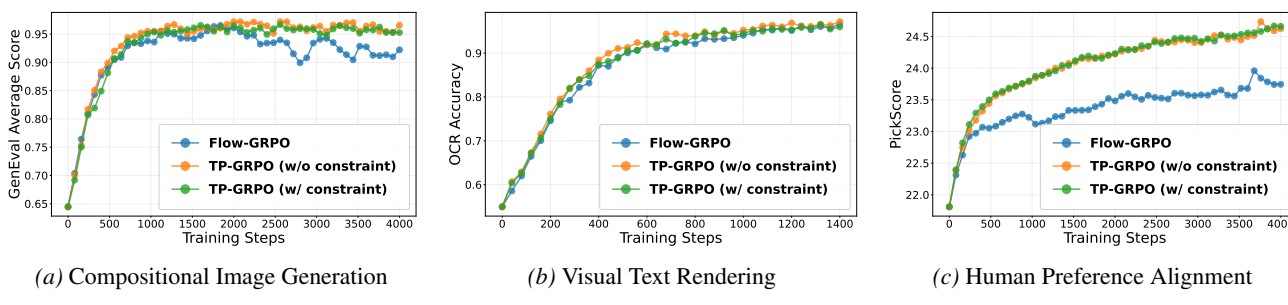

*(a)* Compositional Image Generation     *(b)* Visual Text Rendering     *(c)* Human Preference Alignment

*Figure 4.* Training curves on three evaluation tasks. The two TP-GRPO variants differ in whether they apply the consistency constraint in Definition 5.1.

table, we present the training curves in Figure 4. These curves are obtained from experiments where we remove the KL penalty in Eq. 4 to assess the exploratory capability of our method. In this unconstrained setting, TP-GRPO again achieves superior performance on all three tasks. The gain is especially notable on PickScore, where the unbounded, non–rule-based reward allows our method to better exploit the optimization signal. Our checkpoint at step ∼700 attains a reward comparable to Flow-GRPO at step ∼2300, demonstrating faster convergence and higher performance.

### 6.3. Further Analysis

We further analyze the key components that affect our method's optimization behavior in GRPO. In this section, we remove the KL penalty term in Eq. 4 during training. This setup isolates how the examined factors shape the *pure*

policy optimization dynamics with respect to the reward signal, without confounding effects from regularization.

**Window Size in SDE sampling.** Because we assign rewards at each timestep, the number of steps on which we apply SDE sampling (*i.e.*, the sampling-window size during training) directly affects how the model updates. Figure 6 shows the performance over 2400 training steps as we vary this window. The red curve corresponds to the default configuration, which applies SDE sampling to all steps[1]. Reducing the window size lowers the cost of intermediate sampling and thus shortens training time for a fixed number of epochs. Interestingly, moderately shrinking the window (*e.g.*, to a size of 8) also improves performance. This is consistent with the fact that the final image is largely determined by earlier denoising steps, while the last 1–2 steps have negligible

---

[1]Flow-GRPO default does not apply SDE at the final step.

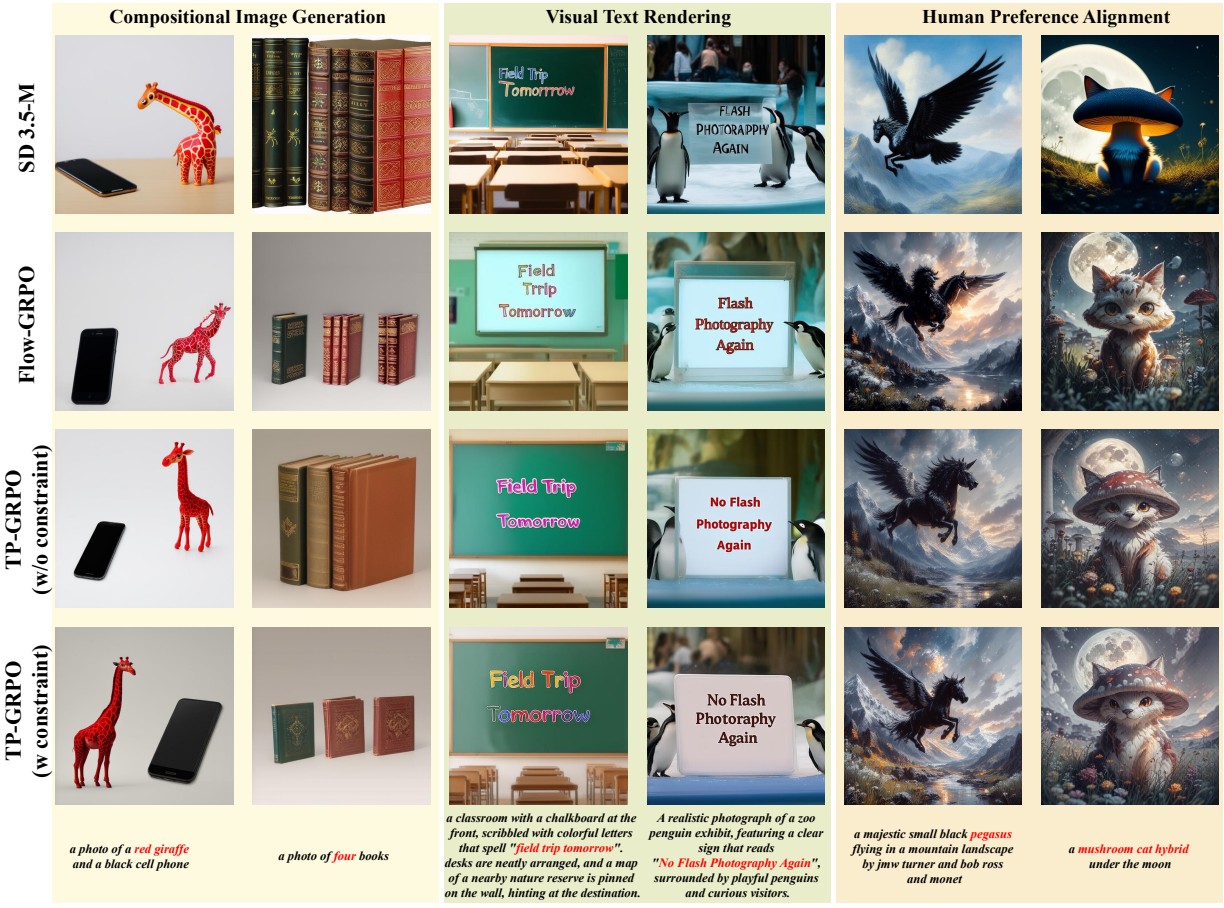

*Figure 5.* Qualitative comparison across three tasks. Compositional Image Generation, Visual Text Rendering, and Human Preference Alignment, respectively, assess color/counting, text rendering, and content alignment (including aesthetics).

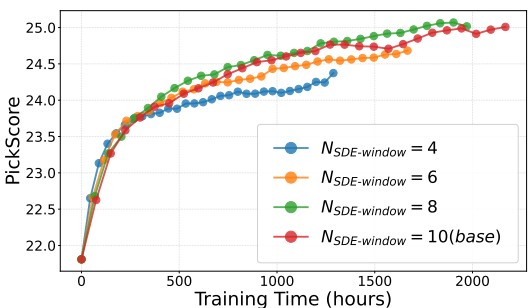

*Figure 6.* Comparison across different numbers of SDE-sampling steps. Performance is reported over 2400 training steps with corresponding training time.

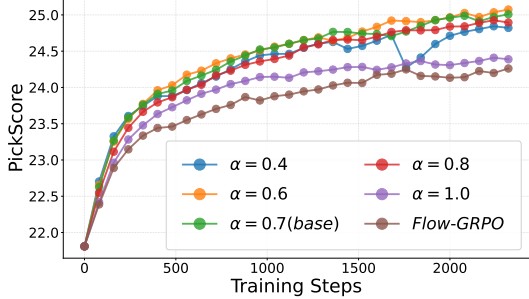

*Figure 7.* Comparison across different noise level $\alpha$. Larger $\alpha$ indicates more stochastic SDE sampling. Within a suitable range, our method consistently surpasses standard Flow-GRPO.

influence and rarely contain turning points. However, when the window is reduced too aggressively (e.g., to 4 steps), performance drops sharply. We attribute this degradation to skipping optimization on later steps and consequently failing to capture turning points that occur in those steps.

**Noise-Scale Choice in the SDE Sampler.** The coefficient $\alpha$ in Eq. 2 scales the stochastic term in the SDE sampler and thereby controls the variability of the denoising trajectory.

Larger $\alpha$ produces more diverse intermediate states. Flow-GRPO uses $\alpha = 0.7$ as its default, and we take this value as the reference in Figure 7, where we report our results under different $\alpha$. Small deviations around $0.7$ do not substantially change the learning trend. When $\alpha$ is too small (*e.g.*, $0.4$), the stochasticity becomes insufficient: the curve exhibits oscillation around $1750$ steps and stays mostly below that of $\alpha = 0.7$. When $\alpha$ is too large (*e.g.*, $1.0$), the intermediate latents become overly diverse, the optimiza-

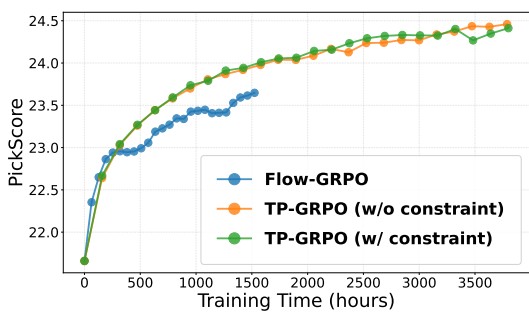

*Figure 8.* Wall-clock time comparison given the same steps.

tion direction becomes unstable, and performance clearly degrades. These results suggest that GRPO requires a balanced level of stochasticity: both too little and too much noise harm learning. Across all tested values, however, our method consistently outperforms the Flow-GRPO baseline, indicating robustness to this hyperparameter.

**Comparison of Performance under the Same Wall-Clock Time.** Our TP-GRPO requires additional sampling to provide the fine-grained signals needed for effective RL optimization. To evaluate whether this overhead is justified, we compare its performance with Flow-GRPO under the same wall-clock training time on the human preference alignment task. As shown in Figure 8, although our method incurs higher time cost per training step, it still consistently outperforms Flow-GRPO when compared at equal wall-clock time. These results indicate that the additional sampling overhead is well justified by the resulting performance gains.

## 7. Conclusion

In this paper, we identified two limitations of existing Flow-based GRPO methods. First, terminal rewards are uniformly propagated to all denoising steps. This causes step-wise reward sparsity and local misalignment. Second, group-wise ranking compares trajectories only at matched timesteps. It ignores within-trajectory dependencies and delayed effects. To address these issues, we proposed TurningPoint-GRPO (TP-GRPO). TP-GRPO uses step-level incremental rewards, which accurately models each denoising action's local effect. Our method also identifies turning points, which are steps that flip the local reward trend. We assign these turning-point actions an aggregated long-term reward to capture their delayed impact. Turning points are detected via sign changes of incremental rewards. This keeps the method efficient and hyperparameter-free. Extensive experiments show that TP-GRPO could effectively improve generations.

## Acknowledgements

This research was supported in part by Zhejiang Provincial Natural Science Foundation of China under Grant LD24F020016, CASIC Guangxin Intelligent Technology, Key R&D Program of Ningbo under Grant 2025Z128, and Alibaba Group through Alibaba Research Intern Program.

We would also like to thank all anonymous reviewers for their insightful comments and helpful suggestions, which have significantly improved the quality of this work.

## Impact Statement

This paper presents work whose goal is to advance the field of Machine Learning. There are many potential societal consequences of our work, none which we feel must be specifically highlighted here.

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

The **Appendix** is structured as follows:

## A. Experimental Results on FLUX.1-dev

To further verify the robustness of our method across different architectures, we adopt FLUX.1-dev (Labs, 2024) as the base model and use PickScore (Kirstain et al., 2023) as the reward model. Following the Flow-GRPO configuration, we set $\alpha$ to 0.8, use $T = 6$ sampling steps during training and $T = 28$ during inference, and choose a group size of $G = 24$. The classifier-free guidance scale is fixed to 3.5. The corresponding training curves are shown in Figure 9. Our method consistently outperforms Flow-GRPO under this setup, demonstrating the effectiveness of the proposed step-level reward design and our mechanism for capturing turning points.

## B. Details of the Experimental Configuration

Our experiments are conducted based on the Flow-GRPO codebase (Liu et al., 2025a). We train all models using 32 NVIDIA H20 GPUs. To maximize performance, we compute advantages on a per-prompt basis[2]. Following the practice of DanceGRPO (Xue et al., 2025), we set the KL penalty coefficient $\beta$ to balance fast convergence and the prevention of reward hacking. Specifically, we use $\beta = 0.0004$ for Compositional Image Generation and Visual Text Rendering, and $\beta = 0.0001$ for Human Preference Alignment. For both training and evaluation, in some cases we adopt newer reward model checkpoints that exhibit stronger performance than the defaults used in Flow-GRPO. The exact reward models and their versions are summarized in Table 2.

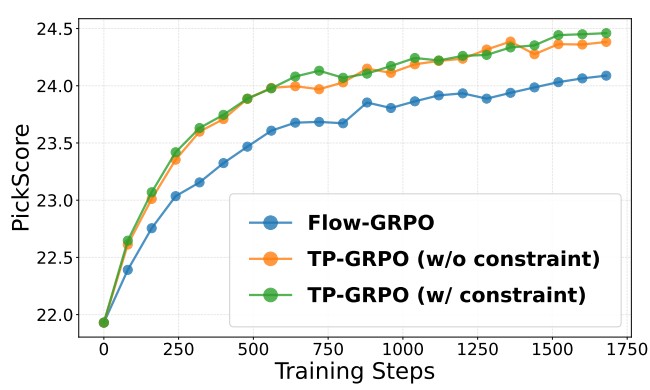

*Figure 9.* Training curves with FLUX.1-dev as base model

*Table 2.* Used reward models and their links.

| Models | Links |
|---|---|
| Aesthetic Score (Radford et al., 2021) | https://github.com/LAION-AI/aesthetic-predictor |
| PickScore (Kirstain et al., 2023) | https://huggingface.co/yuvalkirstain/PickScore_v1 |
| DeQA Score (You et al., 2025) | https://huggingface.co/zhiyuanyou/DeQA-Score-Mix3 |
| ImageReward (Xu et al., 2023) | https://huggingface.co/THUDM/ImageReward |
| UnifiedReward (Wang et al., 2025) | https://huggingface.co/CodeGoat24/UnifiedReward-qwen-7b |
| OCR Accuracy (Cui et al., 2025) | https://github.com/PaddlePaddle/PaddleOCR/tree/release/3.2 |
| GenEval Score (Ghosh et al., 2023) | https://mmdetection.readthedocs.io/en/v2.28.2/ |

---

[2]https://github.com/yifan123/flow_grpo/issues/52

# C. Theoretical Analysis
# of Sign-based Turning-Point Criteria

## C.1. Sign Consistency of Rewards Under Definition 4.1

**Lemma C.1.** *For any turning point selected by Definition 4.1, the sign of its local reward and aggregated long-term reward is the same, i.e., $r_t \cdot r_t^{agg} > 0$.*

*Proof.* By definition,

$$r_t = R\big(\boldsymbol{x}_{t-1}^{\text{ODE}(t-1)}\big) - R\big(\boldsymbol{x}_t^{\text{ODE}(t)}\big), \qquad r_t^{\text{agg}} = R\big(\boldsymbol{x}_0\big) - R\big(\boldsymbol{x}_t^{\text{ODE}(t)}\big).$$

Definition 4.1 states that $t$ is a turning point only if

$$\text{sign}\Big(R(\boldsymbol{x}_{t-1}^{(t-1)}) - R(\boldsymbol{x}_t^{(t)})\Big) \cdot \text{sign}\Big(R(\boldsymbol{x}_0^{(0)}) - R(\boldsymbol{x}_t^{(t)})\Big) > 0.$$

Identifying the terms, we see that

$$\text{sign}\big(r_t\big) \text{sign}\big(r_t^{\text{agg}}\big) > 0.$$

Therefore, $r_t$ and $r_t^{\text{agg}}$ have the same nonzero sign, which implies

$$r_t \cdot r_t^{\text{agg}} > 0.$$

$\square$

## C.2. Sign Consistency and Magnitude of Rewards Under Definition 5.1

### C.2.1. SIGN CONSISTENCY OF LOCAL AND AGGREGATED REWARDS

**Lemma C.2.** *For any turning point selected by Definition 5.1, the sign of its local reward and aggregated long-term reward is the same, i.e., $r_t \cdot r_t^{agg} > 0$.*

*Proof.* Recall the definitions of our computed reward (Eq. 7 and 8)

$$r_t = R\big(\boldsymbol{x}_{t-1}^{(t-1)}\big) - R\big(\boldsymbol{x}_t^{(t)}\big),$$
$$r_t^{\text{agg}} = R\big(\boldsymbol{x}_0^{(0)}\big) - R\big(\boldsymbol{x}_t^{(t)}\big).$$

From Definition 5.1, the only condition we need for this lemma is

$$\text{sign}\big(R(\boldsymbol{x}_{t-1}^{(t-1)}) - R(\boldsymbol{x}_t^{(t)})\big) \cdot \text{sign}\big(R(\boldsymbol{x}_0^{(0)}) - R(\boldsymbol{x}_{t-1}^{(t-1)})\big) > 0. \tag{9}$$

The other constraints in the definition ($s_{t+1} < 0$, $s_t > 0$, and $1 \le t \le T - 1$) are not used in this particular argument and are omitted here for clarity.

Define the shorthand

$$\begin{cases} a := R(\boldsymbol{x}_{t-1}^{(t-1)}), \\ b := R(\boldsymbol{x}_t^{(t)}), \\ c := R(\boldsymbol{x}_0^{(0)}). \end{cases}$$

Then

$$r_t = a - b, \qquad r_t^{\text{agg}} = c - b.$$

In terms of $a, b, c$, the key consistency condition Eq. 9 becomes

$$\text{sign}(a - b) \text{sign}(c - a) > 0. \tag{10}$$

From Eq. 10, the product of the two signs is strictly positive, so they must be equal:

$$\text{sign}(a - b) = \text{sign}(c - a).$$

Hence

$$\text{sign}(r_t) = \text{sign}(a - b) = \text{sign}(c - a). \tag{11}$$

We now show that $\text{sign}(c - b) = \text{sign}(c - a)$, which will imply $\text{sign}(r_t) = \text{sign}(r_t^{\text{agg}})$.

Consider two cases:

**Case 1:** $\text{sign}(a - b) > 0$. Then $a > b$, and by Eq. 10 we also have $\text{sign}(c - a) > 0$, so $c > a$. Consequently,

$$c > a > b \quad \Rightarrow \quad c - b > 0.$$

Thus $\text{sign}(c - b) > 0 = \text{sign}(c - a)$, and, together with Eq. 11, we obtain

$$\text{sign}(r_t) = \text{sign}(r_t^{\text{agg}}).$$

**Case 2:** $\text{sign}(a - b) < 0$. Then $a < b$, and by Eq. 10 we have $\text{sign}(c - a) < 0$, so $c < a$. Consequently,

$$c < a < b \quad \Rightarrow \quad c - b < 0.$$

Thus $\text{sign}(c - b) < 0 = \text{sign}(c - a)$, and again by Eq. 11,

$$\text{sign}(r_t) = \text{sign}(r_t^{\text{agg}}).$$

In both cases, we conclude that

$$\text{sign}(r_t) = \text{sign}(r_t^{\text{agg}}),$$

which implies

$$r_t \cdot r_t^{\text{agg}} > 0,$$

since Eq. 10 enforces strict inequalities and excludes the degenerate zero case. This completes the proof. $\square$

### C.2.2. COMPARISON OF ABSOLUTE VALUES OF LOCAL AND AGGREGATED REWARDS

**Lemma C.3.** *Under the same notation and condition Eq. 9 as in the proof of Lemma C.2, if $r_t \cdot r_t^{agg} > 0$ holds, then*

$$\left| r_t^{agg} \right| > \left| r_t \right|.$$

*Proof.* We use the same shorthand as before:

$$\begin{cases} a := R(\boldsymbol{x}_{t-1}^{(t-1)}), \\ b := R(\boldsymbol{x}_t^{(t)}), \\ c := R(\boldsymbol{x}_0^{(0)}). \end{cases}$$

so that

$$r_t = a - b, \qquad r_t^{\text{agg}} = c - b.$$

From the previous lemma, the condition Eq. 9 implies

$$\text{sign}(r_t) = \text{sign}(r_t^{\text{agg}}),$$

*i.e.*, $r_t$ and $r_t^{\text{agg}}$ have the same sign, and therefore $r_t \cdot r_t^{\text{agg}} > 0$.

To show $|r_t^{\text{agg}}| > |r_t|$, we analyze the two possible sign cases.

**Case 1:** $r_t > 0$ **and** $r_t^{\text{agg}} > 0$**.** In this case,

$$a - b > 0 \quad \text{and} \quad c - b > 0,$$

which means $a > b$ and $c > b$. The consistency condition Eq. 9 gives

$$\text{sign}(a - b)\,\text{sign}(c - a) > 0.$$

Since $a - b > 0$, we must have $c - a > 0$, and hence

$$c > a > b.$$

Therefore,

$$c - b > a - b \quad \Rightarrow \quad r_t^{\text{agg}} > r_t > 0.$$

Taking absolute values preserves the inequality:

$$|r_t^{\text{agg}}| = r_t^{\text{agg}} > r_t = |r_t|.$$

**Case 2:** $r_t < 0$ **and** $r_t^{\text{agg}} < 0$**.** In this case,

$$a - b < 0 \quad \text{and} \quad c - b < 0,$$

which means $a < b$ and $c < b$. Again, from Eq. 9,

$$\text{sign}(a - b)\,\text{sign}(c - a) > 0.$$

Since $a - b < 0$, we must have $c - a < 0$, and hence

$$c < a < b.$$

Therefore,

$$c - b < a - b < 0 \quad \Rightarrow \quad r_t^{\text{agg}} < r_t < 0.$$

Taking absolute values reverses the inequality:

$$|r_t^{\text{agg}}| = -(c - b) > -(a - b) = |r_t|.$$

In both cases, we obtain

$$|r_t^{\text{agg}}| > |r_t|,$$

which proves the claim. □

### C.3. Why Sparse Rewards Are Problematic for Flow-GRPO

The core issue brought by sparse rewards in Flow-GRPO is the mismatch between the Flow Matching model's time-dependent input and its time-invariant reward signal. In Flow-GRPO, the gradient $\nabla_\theta \mathcal{J}(\theta) \approx \mathbb{E}_{p,t}\left[\nabla_\theta \log p_\theta(\boldsymbol{x}_{t-1}|\boldsymbol{x}_t, t) \cdot A(R(\boldsymbol{x}_0))\right]$ uses an advantage $A(R(\boldsymbol{x}_0))$ that is constant across all denoising steps $t$. This results in a non-injective mapping where the model receives identical supervision for diverse denoising actions in a trajectory, ignoring the temporal nature of the velocity field. Our method introduces time-injective reward assignment. By computing step-wise rewards $r_t$ and long-term rewards $r_t^{\text{agg}}$, we transform the advantage into $A_t = A(r_t \text{ or } r_t^{\text{agg}})$, which explicitly incorporates $t$. This restores the injectivity between the policy's conditioning input $t$ and the optimization signal: $p_\theta(\boldsymbol{x}_{t-1}|\boldsymbol{x}_t, t) \rightarrow A_t$. Thus, TP-GRPO allows the policy to isolate the local gain of a denoising action from the global trajectory effect, effectively resolving the misalignment brought by sparse reward.

## D. Balancing Operations for Improved Optimization with TP-GRPO

When deploying TP-GRPO, we also introduce a balancing strategy to control the frequency of reward replacement within each batch. Since $r_t^{\text{agg}}$ can be positive or negative, a batch dominated by negative values may cause the policy to reject most actions, while a batch dominated by positive values may lead to overly conservative updates and insufficient exploration. To prevent either partition from dominating the optimization, we compute the number of positive and negative $r_t^{\text{agg}}$ in each batch and enforce a balanced selection. Specifically, we only keep an equal number of positive and negative samples for replacing $r_t$. The remaining samples are discarded by ranking $|r_t^{\text{agg}}|$ and dropping those with the smallest magnitudes.

# E. Pseudocode

---

**Algorithm 1** TP-GRPO

---

**Require:** Policy (score) model $p_\theta$, reward model $R(\cdot)$, SDE noise level $\alpha$, KL regularization coefficient $\beta$, group size $G$, number of sampling steps $T$, number of training iterations $K$, number of samples per iteration $N$.

  **for** $k = 0$ **to** $K - 1$ **do**
    // **Stage 1: Sample trajectories and collect intermediate rewards**
    **for** $i = 0$ **to** $N - 1$ **do**
      Sample terminal noise $\boldsymbol{x}_{T,i}$ and corresponding prompt $\boldsymbol{c}_i$.
      Run SDE sampling (Eq. 2) from $t = T$ to $t = 0$ to obtain a noisy trajectory $\{\boldsymbol{x}_{t,i}\}_{t=T}^0$.
      For each $t$, run $t$ ODE steps starting from $\boldsymbol{x}_{t,i}$ to obtain the corresponding clean sample $\boldsymbol{x}_{t,i}^{(t)}$.
      Compute intermediate rewards $\{R(\boldsymbol{x}_{t,i}^{(t)})\}_{t=T}^0$.
    **end for**
    // **Stage 2: Optimize $p_\theta$ with TP-GRPO**
    **for** $t = T$ **to** $2$ **do**
      **for** each group $\{\boldsymbol{x}_{t,g}\}_{g=1}^G$ **do**
        **for** each $g \in \{1, \ldots, G\}$ **do**
          Initialize flag `use_aggregated_effect` $\leftarrow$ **False**.
          **if** $t = T$ **and** `select good starting point` **then**
            // **First step: apply Remark 5.2**
            **if** $\text{sign}\big(R(\boldsymbol{x}_{T-1,g}^{(T-1)}) - R(\boldsymbol{x}_{T,g}^{(T)})\big) \cdot \text{sign}\big(R(\boldsymbol{x}_{0,g}^{(0)}) - R(\boldsymbol{x}_{T,g}^{(T)})\big) > 0$ **then**
              `use_aggregated_effect` $\leftarrow$ **True**
            **end if**
          **else**
            // **Later steps: select turning points via Definition 5.1**
            Compute $s_t$ and $s_{t+1}$ according to Eq. 6.
            **if** $s_{t+1} < 0$ **and** $s_t > 0$ **and** $\text{sign}\big(R(\boldsymbol{x}_{t-1,g}^{(t-1)}) - R(\boldsymbol{x}_{t,g}^{(t)})\big) \cdot \text{sign}\big(R(\boldsymbol{x}_{0,g}^{(0)}) - R(\boldsymbol{x}_{t,g}^{(t)})\big) > 0$ **then**
              `use_aggregated_effect` $\leftarrow$ **True**
            **end if**
          **end if**
          **if** `use_aggregated_effect` **then**
            Compute aggregated reward $r_{t,g}^{\text{agg}}$ using Eq. 8.
          **else**
            Compute stepwise reward $r_{t,g}$ using Eq. 7.
          **end if**
        **end for**
      **end for**
      Compute advantages using $r_t$ and $r_t^{\text{agg}}$ (when applicable), according to Eq. 3.
      Update model parameters $\theta$ using the GRPO objective in Eq. 4.
    **end for**
  **end for**

---

# F. More Analysis on Turning Points

In this section, we provide additional discussions and visualizations to illustrate the effects of turning points.

## F.1. Contrast with Credit Assignment Techniques

Credit assignment, *i.e.*, assigning rewards to different steps of a trajectory, is a fundamental challenge in RL. Prior works (Malkin et al., 2022; Madan et al., 2023) provide useful insights, but they typically rely on algebraic constraints such as Trajectory Balance and require an explicit target density. In contrast, our setting considers continuous denoising trajectories optimized with external, non-differentiable image-scoring models, where such structures are unavailable. Our turning-point locating mechanism instead captures trajectory-level structure to provide more informative signals for RL fine-tuning.

### F.2. Subtraction-based Rewards for Normal Points vs. Turning Points

We analyze the subtraction-based rewards $r_t$ and $r_t^{\text{agg}}$ at the intermediate step $t = 6$ as an example in Figure 10. By separately computing the mean rewards for turning points and normal points across training iterations, we observe that their reward distributions are actually distinct. This confirms that our method effectively assigns different reward signals to these two categories of points, tailoring the feedback to the specific role of the action within the trajectory. By integrating both types of rewards into the total reward, our approach provides a more comprehensive objective, enabling it to outperform standard Flow-GRPO.

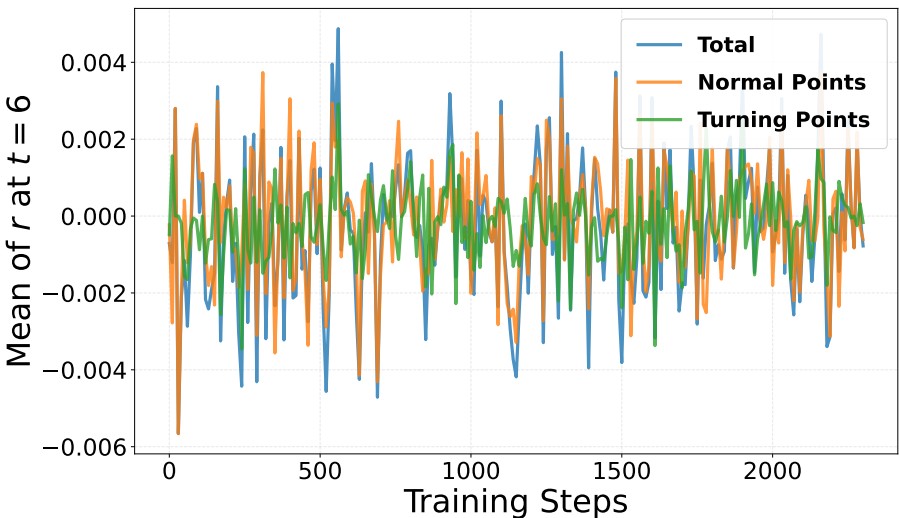

*Figure 10.* Our rewards computed at the intermediate step $t = 6$.

### F.3. Average Count of Good and Bad Turning Points

We tracked the average number of turning points at within a batch of 36 samples in Figure 11, distinguishing between "good" turning points (those guiding the trajectory toward higher rewards) and "bad" turning points (those leading to lower rewards). We observe that the frequency of turning points does not diminish as training progresses, e.g., the **combined count** remains stable at a ratio of approximately 4–6/36 (roughly 0.11–0.16). This ensures a steady stream of informative signals, allowing TP-GRPO to effectively guide the policy throughout the optimization process.

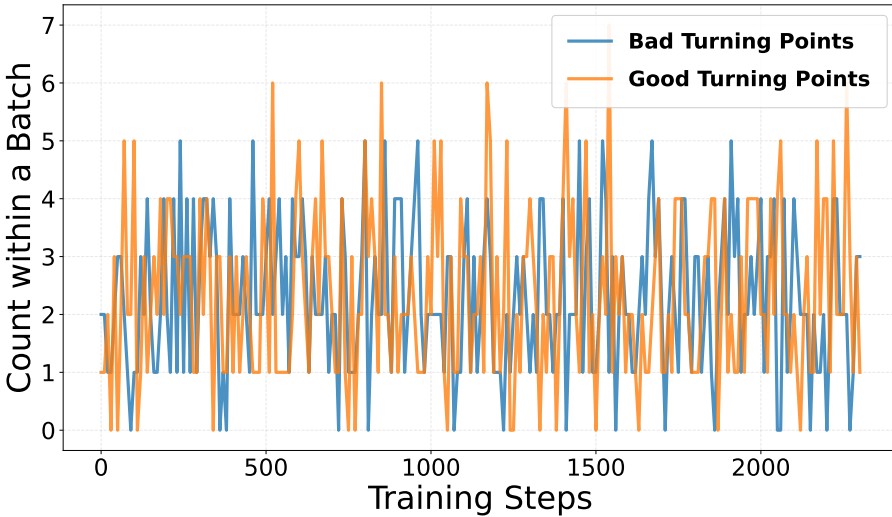

*Figure 11.* The average count of both "good" and "bad" turning points within a batch of 36 samples during the training process.

# G. More Qualitative Comparisons

In this section, we present additional qualitative comparisons on three tasks. Figure 12, Figure 13, and Figure 14 show results on Compositional Image Generation, Visual Text Rendering, and Human Preference Alignment, respectively.

In Compositional Image Generation, we observe that Flow-GRPO sometimes fails to generate accurate images. For example, it produces an unnecessary sandwich for the prompt in the second column, and some generations lose fine details, making them hard to discern (Columns 3 and 5). We attribute this to insufficient reward signals: since the task is rule-driven, the outcome-based reward is inherently sparse, and when the policy model cannot obtain informative process rewards, generation may fail. In contrast, our method provides step-wise rewards, enabling more stable optimization.

For Visual Text Rendering, most samples are rendered well. However, Flow-GRPO occasionally omits short words (*e.g.*, "the" in the fourth column), and some characters overlap, leading to an inaccurate appearance. Our method performs consistently well on this task while preserving the overall aesthetics of the images.

For Human Preference Alignment, our method exhibits strong capability in capturing details and aligning with the prompt (*e.g.*, the "webs" in the fourth column and the "digital art" style in the fifth column). The layout is also more reasonable: for instance, in the second column, our generation separates the city landscape from the planet as implicitly indicated by the prompt, whereas Flow-GRPO's generation confounds these two concepts.

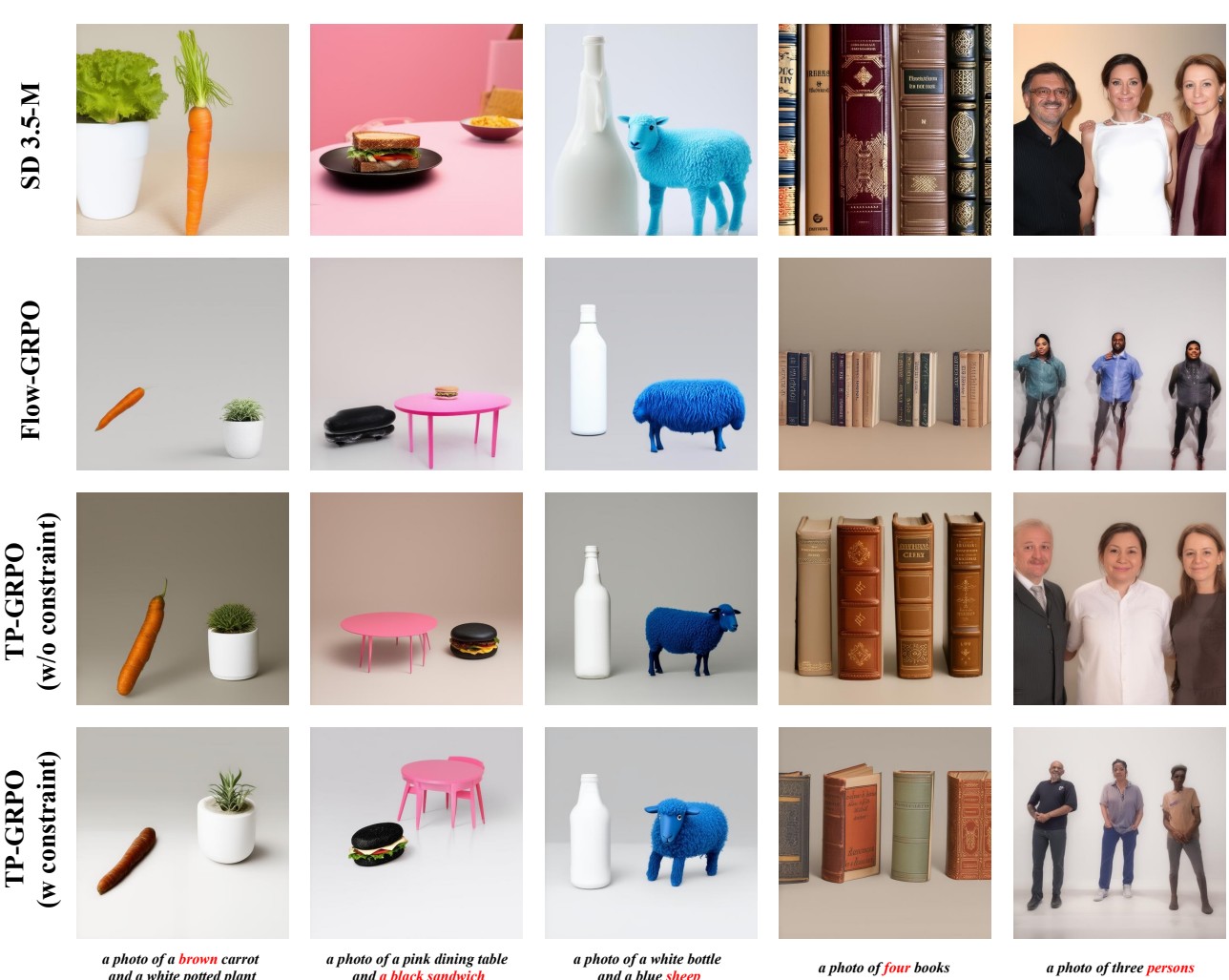

*Figure 12.* Additional qualitative comparison on the Compositional Image Generation task.

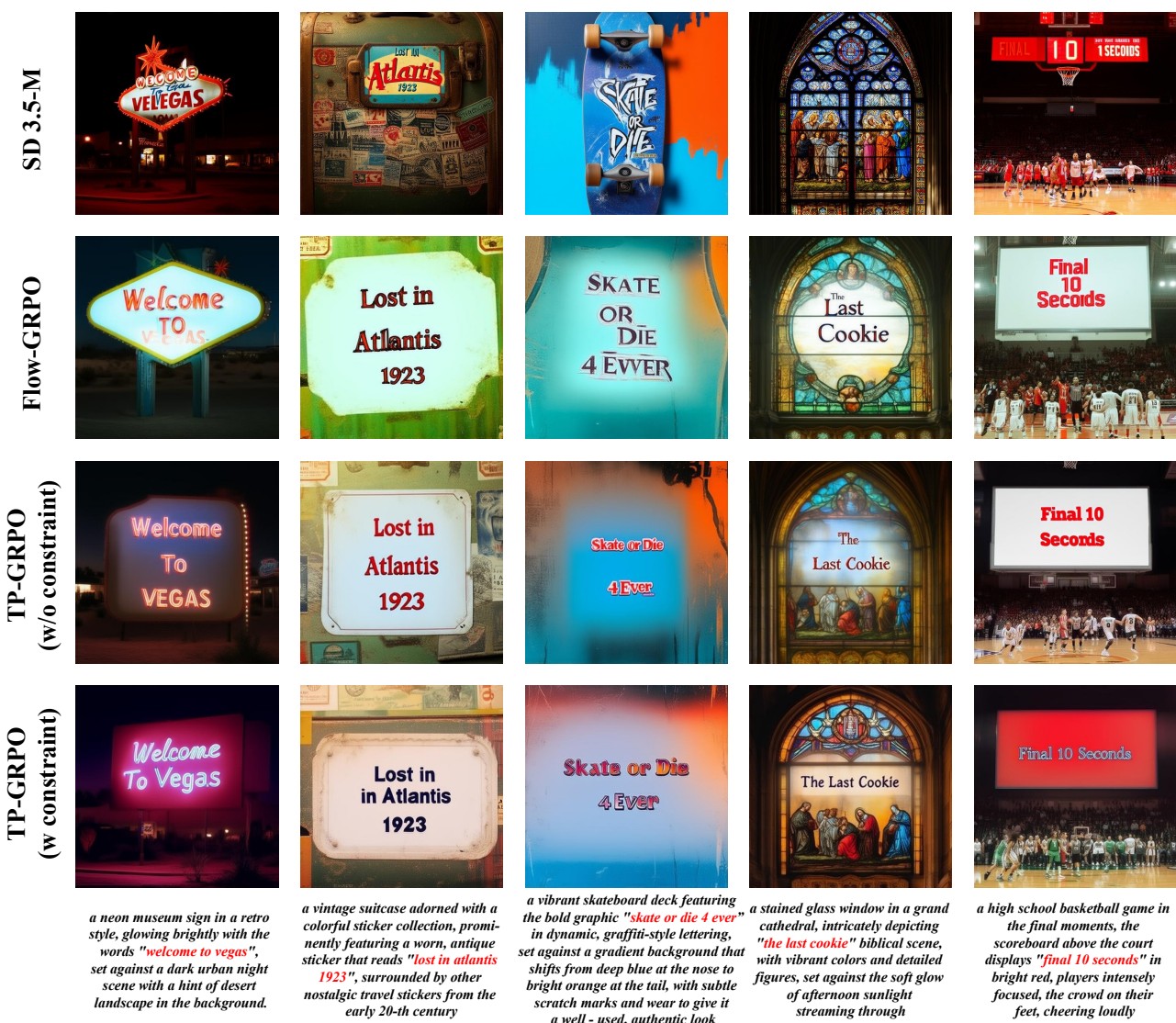

*Figure 13.* Additional qualitative comparison on the Visual Text Rendering task.

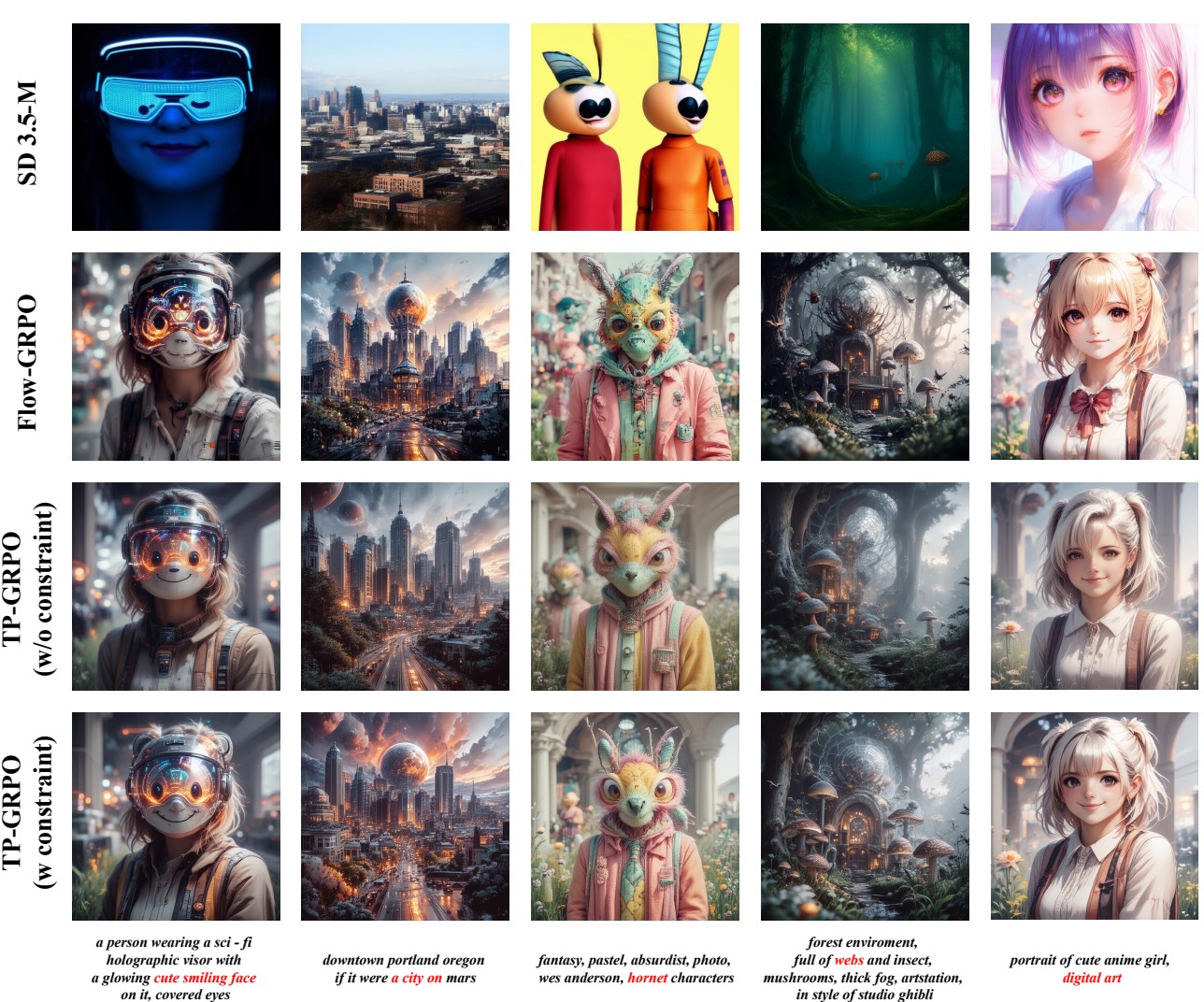

*Figure 14.* Additional qualitative comparison on the Human Preference Alignment task.

