# OpenReview forum: "Alleviating Sparse Rewards by Modeling Step-Wise and Long-Term Sampling Effects in Flow-Based GRPO"
_ICML.cc/2026/Conference — ICML 2026 regular_

### Official Review · Reviewer_dNJw · 2026-03-04

**Soundness:** 3
**Presentation:** 3
**Significance:** 3
**Originality:** 3
**Overall Recommendation:** 5
**Confidence:** 3

**Summary:**

This paper proposes TurningPoint-GRPO (TP-GRPO), a reinforcement learning framework for fine-tuning Flow Matching models. The authors identify two limitations in existing methods like Flow-GRPO: (1) reward sparsity caused by propagating a single terminal reward uniformly to all denoising steps, and (2) the neglect of within-trajectory dependencies. To address these, TP-GRPO introduces step-wise incremental rewards that isolate each step's contribution, and a "turning point" mechanism that detects sign changes in incremental rewards to assign aggregated long-term rewards for delayed credit assignment. The method is evaluated on Compositional Image Generation, Visual Text Rendering, and Human Preference Alignment using SD3.5-M and FLUX.1-dev.

**Compliance With Llm Reviewing Policy:**

Affirmed.

**Final Justification:**

Thank you to the authors for the detailed and thoughtful rebuttal. I appreciate the additional clarifications and newly provided analyses.

Overall, my main concerns regarding Questions 1–3 have been adequately addressed. In particular, the clarification of the sampling procedure (SDE/ODE branching), the additional analysis on turning points, and the justification of computational overhead significantly improve the technical clarity and completeness of the paper.

For Question 4, while the authors provided a reasonable explanation that rewards are computed on fully denoised images and thus avoid noise-level bias, I still believe the formulation and its implications could benefit from clearer formalization in the final version. Nevertheless, the current response is sufficient to alleviate my major concern.

**Soundness.** The method appears technically solid. The rebuttal clarifies the sampling procedure (particularly the SDE/ODE branching) and provides additional analysis on turning points.

**Presentation.** The paper is clear and well organized. The rebuttal further improves clarity by elaborating on implementation details and adding analysis. These clarifications should be incorporated into the final manuscript, especially in Section 4.2 and the description of the turning point mechanism.

**Significance.** The work targets an important issue in RL-based training for diffusion/flow models, namely reward sparsity and credit assignment. The turning point mechanism offers a practical way to capture long-term dependencies, and the empirical results show consistent improvements across tasks. While the gains are moderate, the direction is meaningful and could motivate further research.

**Originality.** There is some overlap between the stepwise reward formulation and concurrent work (e.g., DenseGRPO), and I partially share this concern. However, given that this is contemporaneous work, the turning point mechanism still provides a distinct contribution, particularly in modeling trajectory-level structure. Overall, the level of novelty is acceptable.

Regarding novelty, I agree with the concern that the stepwise reward formulation is related to concurrent work such as [DenseGRPO] (as noted by Reviewer CQzP). However, as this is contemporaneous work, I believe the proposed “turning point” mechanism still provides a distinct and meaningful contribution, particularly in modeling long-term dependencies within trajectories. Therefore, I consider the originality to be acceptable.

Taking all aspects into account, the rebuttal has positively updated my assessment. I am increasing my score (Accept) accordingly, although I will slightly lower my confidence (5->3) due to the overlap with concurrent work.

**Key Questions For Authors:**

1. Please refer to the weaknesses 1,3,4 above.

2. The turning point concept appears potentially transferable to improving high-quality inference in diffusion models. Could the authors further discuss this possibility?

The paper is currently borderline, though I am inclined toward acceptance. If the authors can satisfactorily address the concerns, I would consider further increasing my score. However, if Weakness 4 remains unresolved, I may need to re-evaluate my assessment.

**Limitations:**

yes

**Strengths And Weaknesses:**

**Strengths**
1. The proposed concept of the “turning point” is general and novel. This idea is transferable to other domains in diffusion or flow models.
2. Detecting reward trend reversals via sign changes in incremental rewards is a reasonable design, it avoids the need for additional hyperparameters and is easy to implement on standard flow matching pipelines.
3. Experimental results verify the effectiveness of the proposed method.

**Weaknesses**
1. In my opinion, the contributions lies in the proposed concept of the turning point (appears sufficiently novel). However, regarding dense rewards and their implementation, prior works [1,2,3,4] have already explored these directions extensively. The paper seems to lack a thorough discussion of these related efforts.

2. The turning point concept lacks visualization. I suggest that the authors include a clear illustration of the turning point in Figure 1 to improve clarity and intuition.

3. During optimization, does the gradient at turning points differ from gradients at other steps (e.g., in magnitude)? The experimental section does not provide sufficient analysis on this aspect.

4. Finally, and this is my main concern, I hope the authors can **carefully consider** the following point. In diffusion models, there is a well-known phenomenon in single-step prediction. Predicted results from early (high-noise) steps tend to produce statistically lower rewards than those from later (low-noise) steps. *I am quite confident about this empirical pattern.* Hence, the formulation in Eq. (7) may inherently introduce bias, i.e., the reward of $r_t$ usually is  larger than 0. In particular, in Remark 5.2, the sign of  $R(x^{(0)}_0) - R(x^{(T)}_T)$  may frequently be negative. What impact could this bias have on the experimental results? Was this issue considered in the method design? If not, how should it be addressed?


[1] Aesthetic Post-Training Diffusion Models from Generic Preferences with Step-by-step Preference Optimization

[2] EasyTune: Efficient Step-Aware Fine-Tuning for Diffusion-Based Motion Generation

[3] Aligning Few-Step Diffusion Models with Dense Reward Difference Learning

[4] TempFlow-GRPO: When Timing Matters for GRPO in Flow Models

---

> ### Author Rebuttal · Authors · 2026-03-31
>
> Thanks for your insightful reviews. Please see our responses below.
>
> [W1] We agree that dense rewards are crucial for diffusion model training. Existing methods (e.g., the papers you referenced) typically model process rewards by isolating factors affected by untargeted steps, ensuring the terminal signal is dominated by the target denoising step. Consequently, the reward of the terminal state in a trajectory could serve as a proxy for the process reward, widely adopting a "sum"-based form. In contrast, our approach utilizes a "subtraction" form (Eq. 7 and 8). By measuring the specific impact of modifying a local sampling step, TP-GRPO computes process rewards independently of the trajectory's terminal state (except when computing rewards for turning points, where terminal reward is essential). Both approaches offer valid process supervision. We will include a discussion of **all your referenced works** in the revised manuscript.
>
> [W2] Thank you for the advice. We will refine Fig. 1 for better clarity in the next version of our paper.
>
> [W3] Theoretically, **there is no fixed magnitude ranking between gradients at turning points and other points, although they are of the same order of magnitude**. We analyze this via the gradient: $\nabla _\theta \mathcal{J}(\theta) \approx \mathbb{E} _{p} [\nabla _\theta \log p _\theta(x _{t-1}|x _t) \cdot A(r)]$. Ignoring constant factors, its magnitude is determined by two terms:
>
> 1. **Denoising Action**: The term $\nabla _\theta \log p _\theta(x _{t-1}| x _t)$ is determined solely by the denoising action and is independent of our reward design.
> 2. **Advantage $A(r)$**: Although $r _t$ (for common points) and $r _t^{\text{agg}}$ (for turning points) are computed differently, they are designed with the same scale: both represent the difference between clean images' rewards induced by different sampling actions. Thus, they are of the same order of magnitude, i.e., $r _t = \Theta(r _t ^\text{agg})$, but there is no consistent ordering between them. (When using the consistency constraint in Definition 5.1 to identify turning points, we have $|r_t^{\text{agg}}| \ge |r_t|$ for rewards computed on turning points. Please see Appendix C.2.2 for the proof. However, note that this inequality holds only for the reward computation at the same point, not across different points).
>
> In summary, gradients at turning points and common points are of the same order of magnitude but do not have a fixed ranking. This is consistent with our design: since the input to the Flow Matching model is strictly same-scaled at all points, the gradients are also same-scaled.
>
> [W4] Thank you for this detailed question. We would like to clarify two points:
>
> 1. **No Noise-Level Misalignment:** We agree that single-step denoising at high noise levels yields noisier outputs than at low noise levels. However, this does not affect our rewards because we compute them exclusively on **fully-denoised clean images, not intermediate latents**. In Eq. 7, $x _{t}^{\text{ODE}(t)}$ represents the clean image after $T-t$ SDE steps and $t$ ODE steps, while $x _{t-1}^{\text{ODE}(t-1)}$ follows $T-t+1$ SDE steps and $t-1$ ODE steps. Both undergo exactly $T$ sampling steps. By deploying the reward model $R(\cdot)$ on fully-denoised results rather than noisy latents, $r _t$ isolates the pure effect of the denoising action at time $t$ without interference from different noise levels. Thus, there is no asymmetric bias.
> 2. **The Sign of Computed Difference:** The rewards $r _t$ and $r _t^{\text{agg}}$ are not statistically biased toward negative values. Since SDE and ODE sampling maintain the same marginal distribution at any given timestep, the sign of $r_t$ depends solely on the quality of the denoising action at $t$, making both positive and negative signs possible. This also applies to the difference $R(x _0^{(0)}) - R(x _T^{(T)})$ in Remark 5.2, which denotes the reward gap between a fully SDE-sampled image and an ODE-sampled image. Its sign is only affected by sampling quality rather than noise-level misalignment.
>
> In summary, **our design does not introduce an inherent bias in the optimization process**. It strictly compares images of the same magnitude and effectively models both short- and long-term effects during denoising.
>
> [Q1] Please see W[1, 3, 4].
>
> [Q2] Thank you for the suggestion. Our definition of turning points is based on the observation that certain pivotal denoising steps dictate the trajectory's reward trend. Integrating this into RL training is natural, as these points provide high-value information for optimization. Regarding inference, a promising approach is to incorporate a "look-ahead" mechanism at these points to adaptively adjust guidance scales, ensuring that high-impact steps receive stronger aesthetic or alignment control. We believe turning points offer a concrete path for future research and will explore this inference-time scaling in subsequent work. :)

---

> > ### Author Rebuttal · Reviewer_dNJw · 2026-04-01
> >
> > Thank you for the authors’ clarification. The concerns regarding Questions 1–3 have been resolved.
> >
> > Regarding Question 4, I still have some remaining issues. In particular, the way intermediate states are obtained in this work needs a clearer formalization, as the description in Section 4.2 is insufficient. I would like to know how the SDE and ODE sampling trajectories are obtained in this work. For $x_T$, the entire trajectory of $v_t$ ($t=0,\dots,T$) is predicted and stored to obtain the SDE trajectory.
> >
> > 1. Then, the stored $v_t$ is reused to obtain the ODE trajectory;
> > 2. Or, for each intermediate state, $v_t$ is re-predicted to obtain the ODE trajectory.
> >
> > If it is case 1, then for the last $t'$ steps of the ODE trajectory, $v_t'=Model(x^{\text{SDE}}_{t'+1})$ rather than $v_t' = Model(x^{ODE}_{t'+1})$. How is the accuracy of the ODE trajectory guaranteed in this case? If it is case 2, is the large amount of computation necessary?
> >
> > Moreover, considering that the authors are currently unable to share qualitative results regarding the turning point, I suggest that the authors provide some brief quantitative results, e.g., how the reward scores change at the turning point, statistically.
> >
> > Will the authors release the code?

---

> > > ### Author Response · Authors · 2026-04-03
> > >
> > > We would like to thank you for the detailed reviews. We are glad that our past responses have addressed the majority of your concerns. Please see our new responses below.
> > >
> > > [Q1] **Regarding the formulation of our sampling strategy**: Thanks for proposing this issue detailedly. We will in turn respond to your concern.
> > >
> > > 1. **Clarification on ODE/SDE Sampling:** Your second interpretation is correct. For an intermediate state $x _t$ reached via SDE denoising from $T$ to $t$, we deploy an ODE sampler for its remaining denoising actions from $t$ to $0$. The update rules of the sampler are defined as:
> > >
> > >    - ODE: $x _{t+\Delta t} = x _t + v _\theta(x _t, t)\Delta t$
> > >    - SDE: $x _{t+ \Delta t} = x _t + [ v _{\theta}(x _t, t) + \frac{\sigma _t^2}{2t}\bigl(x _t + (1-t)v _{\theta}(x _t,t)\bigr)]\Delta t + \sigma _t\sqrt{\Delta t}\,\epsilon$
> > >
> > >    As these samplers maintain the same marginal distribution at each noise level [1], they operate as independent branches from the same noisy latent $x _t$. We utilize the ODE branch at any intermediate step to generate a clean image for reward evaluation. We will refine the formalization in Section 4.2 to clarify this process.
> > >
> > > 2. **Justification of Computational Cost:** Your observation is correct: our method requires more sampling to provide the fine-grained signals necessary for effective RL optimization. To justify this overhead, we compared our performance against Flow-GRPO under identical training time on the human preference alignment task (https://anonymous.4open.science/r/ICML26-rebuttal-C258/toReviewerdNJw.md). The figure demonstrates that while our method requires more time per training step, it surpasses Flow-GRPO under equal wall-clock time. Thus, the extra sampling overhead is effectively justified by the significant improvements in final performance.
> > >
> > >
> > >
> > >
> > >
> > > [Q2] **Regarding providing more analysis on turning points**: Thank you for your suggestions. We have provided additional visualizations to illustrate the effects of turning points at https://anonymous.4open.science/r/ICML26-rebuttal-C258/toReviewerdNJw.md. Our analysis is as follows:
> > >
> > > 1. **Subtraction-based rewards $r$ for normal points vs. turning points**: We analyze the subtraction-based rewards $r _t$ and $r _t^\text{agg}$ at the intermediate step $t=6$ as an example. By separately computing the mean rewards for turning points and normal points across training iterations, we observe that their reward distributions are actually distinct. This confirms that our method effectively assigns different reward signals to these two categories of points, tailoring the feedback to the specific role of the action within the trajectory. By integrating both types of rewards into the total reward, our approach provides a more comprehensive objective, enabling it to outperform standard Flow-GRPO.
> > > 2. **Average count of good and bad turning points**: We tracked the average number of turning points at within a batch of 36 samples, distinguishing between "good" turning points (those guiding the trajectory toward higher rewards) and "bad" turning points (those leading to lower rewards). We observe that the frequency of turning points does not diminish as training progresses, e.g., the **combined count** remains stable at a ratio of approximately 4–6/36 (roughly 0.11–0.16). This ensures a steady stream of informative signals, allowing TP-GRPO to effectively guide the policy throughout the optimization process.
> > >
> > > Our definition of turning point inherently renders it a dynamic concept. For instance, a denoising action identified as a turning point in an early training epoch may no longer fulfill the definition after model parameters are updated, effectively transitioning into a normal point. This makes it impractical to track the turning point behavior of **specific samples** over training time. To provide deeper insight into this mechanism, we focus our analysis on **the overall reward trends and the sustained presence of turning points**, as demonstrated in the linked figures. We will integrate these additional results and the corresponding analysis into the appendix of the final version.
> > >
> > >
> > >
> > >
> > >
> > >
> > >
> > > [Q3] **Regarding Code Availability**: Certainly. Our current implementation is available at https://anonymous.4open.science/r/18EF/README.md. We are committed to publicly releasing the full codebase upon publication.
> > >
> > >
> > >
> > > ---
> > >
> > > [1] Song, Yang, et al. "Score-based generative modeling through stochastic differential equations." *arXiv:2011.13456* (2020).
> > >
> > >
> > >
> > > ---
> > >
> > > ### **Update (following your score update)**
> > >
> > > We would like to thank you for your constructive insights and detailed suggestions throughout this review process. We sincerely appreciate your support and the increase in our rating. We will incorporate all your suggestions and the additional discussion into the final version of our paper. Should you have any further questions or suggestions, we are more than happy to engage in discussions and make any necessary improvements. :)

---

### Official Review · Reviewer_CQzP · 2026-03-10

**Soundness:** 3
**Presentation:** 3
**Significance:** 2
**Originality:** 2
**Overall Recommendation:** 4
**Confidence:** 3

**Summary:**

This paper proposes TurningPoint-GRPO (TP-GRPO) for text-to-image generation. TP-GRPO replaces outcome-based rewards with step-level incremental rewards and identifies turning points to assign aggregated long-term rewards to address limitations of Flow-based GRPO. The comparative experiment with Flow-GRPO verified the effectiveness of the proposed TP-GRPO.

**Compliance With Llm Reviewing Policy:**

Affirmed.

**Final Justification:**

Although the baseline method reproduced by the authors is lower than the results reported in official papers (such as TempFlowGRPO), given the lack of publicly available weights and limited time, I believe this error does not affect the paper's overall completeness. Therefore, I am willing to raise the score to 4. I suggest that the authors make further revisions to the final version as promised.

**Key Questions For Authors:**

1. The novelty of this work needs further verification. The stepwise reward proposed in the paper looks very similar to the method in [4]. Although it is not necessary to compare with contemporary work, could the authors explain whether there is any essential difference between the two?

[4] Deng, H., Yan, K., Mao, C., Wang, X., Liu, Y., Gao, C., & Sang, N. (2026). Densegrpo: From sparse to dense reward for flow matching model alignment. arXiv preprint arXiv:2601.20218.

**Limitations:**

yes

**Strengths And Weaknesses:**

Strengths:

1. The paper is well-structured and well-written.

2. The experimental results verify the effectiveness of the proposed TP-GRPO.

Weaknesses:

1. Several works have attempted to improve upon GRPO in terms of step-wise rewards and turning points, for example, in the field of LLM reasoning. The paper needs to discuss this further in the related work.

2. To verify the effectiveness of TP-GRPO, more comparisons with other methods are necessary, such as [1]-[3].

3. More qualitative comparisons with other methods, such as [1]-[3] or other text-to-image generation methods, are needed.

4. In Figure 4(b), the advantages of TP-GRPO are not obvious. Could you analyze the reasons with examples to verify the generalization ability of the method?

[1] He, X., Fu, S., Zhao, Y., Li, W., Yang, J., Yin, D., ... & Zhang, B. (2025). Tempflow-grpo: When timing matters for grpo in flow models. arXiv preprint arXiv:2508.04324.

[2] Wang, J., Liang, J., Liu, J., Liu, H., Liu, G., Zheng, J., ... & Liang, X. (2025). Grpo-guard: Mitigating implicit over-optimization in flow matching via regulated clipping. arXiv preprint arXiv:2510.22319.

[3] Xue, Z., Wu, J., Gao, Y., Kong, F., Zhu, L., Chen, M., ... & Luo, P. (2025). Dancegrpo: Unleashing grpo on visual generation. arXiv preprint arXiv:2505.07818.

---

> ### Author Rebuttal · Authors · 2026-03-31
>
> Thank you for your detailed feedback. Please see our responses below.
>
> [W1] We will expand the Related Work section in the next version to include more comprehensive discussions on LLMs.
>
> [W2-W3] Per your request on qualitative comparisons with the suggested works, we have provided them at https://anonymous.4open.science/r/ICML26-rebuttal-C258. Please see our analysis:
>
> 1. Tempflow-GRPO and GRPO-Guard: We clarify that TP-GRPO is designed to alleviate sparse rewards and is orthogonal to many techniques in these two works. For instance, GRPO-Guard addresses over-optimization—a goal distinct from our focus on reward sparsity—meaning these approaches are complementary rather than direct alternatives. To ensure a fair evaluation, we trained these models from scratch (as no open-source model checkpoints exist) under identical configurations on human preference alignment task. Our results demonstrate that TP-GRPO consistently achieves superior performance compared to both methods. Specifically, while Tempflow-GRPO adopts a "sum" form to let terminal rewards represent effects for specific steps, our method employs a "subtraction" between two rewards (Eq. 7 and 8) to more directly and accurately isolate local effects from a denoising trajectory. Both are reasonable for modeling process rewards, while our results show that our subtraction-based approach models local effects more purely, leading to better generations.
> 2. DanceGRPO: The community considers both DanceGRPO and Flow-GRPO as Flow Matching-based frameworks with similar backbones. By validating our method on Flow-GRPO—a more widely used baseline—we have already demonstrated the generalization capability of TP-GRPO. Thus, adding our method to DanceGRPO would not provide significantly new insights at this stage, particularly limited rebuttal time and intensive compute required by training. We will, however, deploy our method on DanceGRPO and include those experiments in the next version of our paper. :)
>
> Given the limited rebuttal window and the intensive compute required for full-scale training, we cannot provide all these additional results immediately. However, we will incorporate further discussion on these combinations and include the new experimental results in the appendix of our revised manuscript. :)
>
> [W4] Thanks for your detailed observation. The marginal improvement on the visual text rendering task stems from the task's low difficulty and limited room for enhancement. Here, the rule-based reward only checks for text presence, ignoring image quality or aesthetics. With a maximum score of 1.0, the baseline Flow-GRPO already achieves ~0.96 when fully trained. Because both Flow-GRPO and our method reach the 1.0 ceiling in most cases, performance differences are confined to a tiny fraction of samples. The table below shows the ratio of samples achieving a score of 1.00 on the text rendering task for both Flow-GRPO and our method.
>
> | Method                   | Ratio |
> | ------------------------ | ----- |
> | TP-GRPO (w/o constraint) | 0.665 |
> | TP-GRPO (w constraint)   | 0.674 |
>
> The high ratio confirms that "easy" samples dominate the test set, which masks potential gains. This "ceiling effect" is common in this task: notably, the GRPO-Guard paper you referenced also shows marginal gains over standard Flow-GRPO. In contrast, the human preference alignment task is more difficult, where Figure 4c clearly shows the effectiveness of our method.
>
> [Q1] Thank you for the recommendation. Regarding the similarity to contemporary work you referenced, we would like to clarify that our stepwise reward (Section 5.1) is only a sub-component of our framework. The essential difference—and our primary novelty—lies in the definition of **turning points** (Section 4.2) and the modeling of **aggregation-based long-term effects** (Sections 5.2–5.3). Unlike DenseGRPO, which focuses on local reward assignment, our method addresses implicit interactions across the entire flow trajectory. These core concepts distinguish TP-GRPO from existing methods and provide new insights into flow-based GRPO. We will further clarify these differences and include a detailed discussion of the suggested works in our revised manuscript. :)

---

> > ### Author Rebuttal · Reviewer_CQzP · 2026-04-02
> >
> > 1. The paper may require substantial modification. First, the current title and abstract position step-wise rewards as a primary novelty, which are similar to current works. Second, the paper lacks a sufficiently comprehensive discussion of related methods.
> >
> > 2. I fully understand that the authors have limited computational resources and rebuttal time. Therefore, I will further refer to other reviewers’ comments on the overall completeness and novelty of the work before making my final judgment.

---

> > > ### Author Response · Authors · 2026-04-06
> > >
> > > Thanks for your further response. To ensure all your concerns could be addressed, we have summarized your primary points and our corresponding responses as follows:
> > >
> > > **[Q1] Regarding the novelty of our contribution and the framing of our abstract**
> > >
> > > We appreciate your feedback regarding the positioning of our work. To clarify, our method is two-fold, and both components offer advancements:
> > >
> > > - **Computation of Step-wise Short-term Rewards:** While step-wise reward estimation is a trending topic, our approach is technically distinct from pre-existing methods you referenced. For instance, TempFlowGRPO derives step-wise signals by ensuring the terminal reward is dominated by the target denoising step, which essentially relies on a "sum-based" terminal reward. In contrast, our method employs a "subtraction-based" computation, which isolates local effects by calculating the difference between the rewards obtained before and after a specific denoising action. While both provide valid process rewards, our mechanism adopts a distinct technical path, and we maintain that detailing this computation is essential.
> > >
> > > - **Novel Framework for Identifying Turning Points and Modeling Long-term Rewards:** We identify the phenomenon where specific denoising actions locally alter reward trends and are indicative of the final outcome. This motivates us to leverage the delayed effects of these "turning points" as early signals. **To our knowledge, we are the first to formalize this issue and propose a corresponding framework to model these long-term delayed effects.** This explicitly fills a gap in the current literature. Our manuscript allocates substantial space to this framework, as it provides a comprehensive solution for long-term interaction modeling that has not been addressed in prior work.
> > >
> > > We believe the manuscript presents a strong and novel contribution. Nevertheless, to ensure these two distinct facets of our work are even more clearly highlighted for the reader without relevant knowledge, we will refine the abstract to better reflect our comprehensive framework and expand our discussion on related works in the final version as you suggested. :)
> > >
> > >
> > >
> > > **[Q2] Regarding additional experimental comparisons**
> > >
> > > We appreciate your understanding of the time constraints involved in conducting further experiments. Regarding your initial request for "qualitative" results, we provided a comparison in our previous response. To provide a more detailed performance analysis, we have additionally conducted quantitative evaluations against the baselines you suggested. To ensure a strictly fair evaluation, we trained these models from scratch (as no open-source model checkpoints exist) under identical configurations on the human preference alignment task. The results are summarized in the table below:
> > >
> > >
> > >
> > > |                          | Main Task Metric | Image Quality |       | Preference Score |           |        |
> > > | ------------------------ | ---------------- | ------------- | ----- | ---------------- | --------- | ------ |
> > > |                          | PickScore        | Aesthetic     | DeQA  | ImgRwd           | PickScore | UniRwd |
> > > | Flow-GRPO                | 24.02            | 6.231         | 3.966 | 1.3875           | 24.10     | 3.605  |
> > > | GRPO-Guard               | 23.45            | 6.211         | 3.552 | 1.2637           | 23.72     | 3.475  |
> > > | TempFlowGRPO             | 24.45            | 6.300         | 3.855 | 1.3799           | 24.25     | 3.532  |
> > > | TP-GRPO (w/o constraint) | 24.73            | 6.293         | 3.961 | 1.3714           | 24.46     | 3.600  |
> > > | TP-GRPO (w constraint)   | 24.67            | 6.321         | 3.993 | 1.4419           | 24.61     | 3.640  |
> > >
> > >
> > >
> > > The results demonstrate that our method consistently achieves superior RL optimization compared to all baselines (Column 2). Regarding the aesthetic and generalization tests (Column 3-4 and 5-7), while TempFlowGRPO exhibits competitive performance under specific evaluation protocols, our method consistently achieves better results across the majority of the testing settings. These findings further corroborate the effectiveness and robustness of our TP-GRPO.

---

### Official Review · Reviewer_wofg · 2026-03-11

**Soundness:** 2
**Presentation:** 3
**Significance:** 2
**Originality:** 2
**Overall Recommendation:** 5
**Confidence:** 3

**Summary:**

The authors propose TurningPoint-GRPO (TP-GRPO), a method to provide better step-wise, dense rewards to flow matching models instead of a single terminal reward from the outcome. To do so, at each step, the algorithm completes the image using ODE sampling and compares the reward of the completed image before and after the step to obtain a dense signal.

In addition, TP-GRPO attempts to identify the long term effect of intermediate denoising steps (and provide adequate credit assignment) through turning points. Turning points are defined as steps where the dense reward switches sign in a way consistent with the global reward trend of the denoising (i.e. whether the final image reward is greater/lower than the initial image reward).

The authors test their method on SD3.5-M, performing LoRA finetuning on 3 tasks and comparing with a baseline of Flow-GRPO. They find that TP-GRPO yields slightly better results in all three tasks and improves qualitatively over the baseline. They also ablate the sampling window size and the noise levels.

**Compliance With Llm Reviewing Policy:**

Affirmed.

**Final Justification:**

My main concerns have been addressed. My only remaining suggested improvements are testing the method on additional non-saturated benchmarks and a more thorough investigation of turning points/potential variants of turning points. I would also suggest clarifying the writing to explain the step-wise rewards and turning points better.

**Key Questions For Authors:**

- How much more computationally expensive is this method compared to Flow-GRPO? It seems like you need to do $t$ times more generations to get the denser reward signal.
- Why is the impact more significant for human preference alignment?
- How were the images in the qualitative evaluation selected?
- Have you performed any smaller experiments/analysis that specifically show that the terminal rewards/lack of good credit assignment are a problem for flow-GRPO? And showing how your method helps?

**Limitations:**

yes

**Strengths And Weaknesses:**

**Strengths**
- The proposed method is novel as a variation of flow GRPO.
- The paper is well-written and easy to follow.
- The paper experiments with a large model, evaluates multiple metrics and ablates the impact of window size and noise scale.

**Weaknesses**
- While the application to flow matching GRPO is novel, the issues of terminal/sparse rewards and credit assignment are well-known in the RL literature and the paper would strongly benefit from additional discussion and comparison with existing techniques in RL for dealing with terminal rewards.
  - In fact, the setup resembles GFlowNets where trajectory balance/subtrajectory balance losses have been used successfully to deal with this issue (see [1, 2]).
- The improvements from the method seem marginal for the additional computational cost.
- The paper doesn't convincingly validate that the sparse rewards/credit assignment are a major problem for Flow-GRPO.
- The motivation behind using turning points is unclear to me.
- The paper repeats itself at times (e.g. it is mentioned 4 times that the ODE preserves the same marginal as the SDE).

Ultimately, this method feels like the equivalent of doing a Monte Carlo rollout at each step and comparing the reward before and after.
To increase my score, the paper would need to do a better job of theoretically showing why these local rewards (and the turning point credit assignment) provide better signal (in a way that is worth the additional cost).

[1] Malkin, Nikolay, et al. "Trajectory balance: Improved credit assignment in gflownets." Advances in Neural Information Processing Systems 35 (2022): 5955-5967.
[2] Madan, Kanika, et al. "Learning gflownets from partial episodes for improved convergence and stability." International Conference on Machine Learning. PMLR, 2023.

---

> ### Author Rebuttal · Authors · 2026-03-31
>
> Thanks for your detailed reviews. Please see our responses to both weaknesses and questions.
>
> [W1]  We thank the reviewer for suggesting the GFlowNet papers. While these works offer valuable insights into credit assignment, they are fundamentally inapplicable to our RL optimization for image generation. These methods rely on algebraic constraints (e.g., Trajectory Balance) that require an explicit target density to be matched, whereas our task involves optimization of continuous denoising trajectories using external, non-differentiable image-scoring models where such algebraic structures do not exist. We will clarify these conceptual differences in the revised paper.
>
> [W2] We address this concern in two parts:
>
> - **On performance gains:** Our experiments follow the Flow-GRPO setting. In the compositional image generation and text rendering tasks, the baseline already nears the theoretical ceiling (~0.96/1.00); thus, **significant gains are constrained by the task's limited headroom**. This aligns with prior work [1, 2]. Our method's effectiveness is better reflected in the more challenging human preference task, where there is enough room for gain.
> - **On computation cost:** We provide a comparison of performance over training time in https://anonymous.4open.science/r/ICML26-rebuttal-C258. While our method requires more time per training step, it surpasses Flow-GRPO under equal wall-clock time. Thus, the extra sampling cost is justified by our superior performance.
>
> [W3] The core issue brought by sparse rewards in Flow-GRPO is the mismatch between the Flow Matching model's time-dependent input and its time-invariant reward signal. In Flow-GRPO, the gradient $\nabla_\theta \mathcal{J}(\theta) \approx \mathbb{E} _{p,t} \left[  \nabla _\theta \log p _\theta(x _{t-1}|x _t, t) \cdot A(R(x _0)) \right]$ uses an advantage $A(R(x _0))$ that is constant across all denoising steps $t$. **This results in a non-injective mapping where the model receives identical supervision for diverse denoising actions in a trajectory, ignoring the temporal nature of the velocity field.** Our method introduces time-injective reward assignment. By computing step-wise rewards $r _t$ and long-term rewards $r _t^{\text{agg}}$, we transform the advantage into $A _t = A(r _t \text{ or } r _t ^{\text{agg}})$, which explicitly incorporates $t$. This restores the injectivity between the policy's conditioning input $t$ and the optimization signal: $p _\theta(x _{t-1}|x _t, t) \to A _t$. Thus, TP-GRPO allows the policy to isolate the local gain of a denoising action from the global trajectory effect, effectively resolving the misalignment brought by sparse reward.
>
> [W4] Turning points identify denoising actions that locally alter reward trends, acting as leading indicators for the final reward. In denoising trajectories, intermediate rewards are often oscillatory (Figure 1). A single action can exert a long-term effect that outcome-based rewards in Flow-GRPO cannot capture. By identifying these actions, we use their delayed effects as early signals, allowing the model to anticipate reward trends. Our sign-based criterion and subtraction-based rewards model the specific effects of these points, enabling the policy to prioritize actions that offer sustainable gains.
>
> [W5] Thanks for your advice. We will revise it in the next version.
>
> [Comment] Please see W3 for our theoretical analysis of sparse rewards.
>
> [Q1] While our method requires more sampling, it performs better within the same wall-clock time, as detailed in W2.
>
> [Q2] As noted in W2, human preference alignment task is harder than rule-based tasks. While the latter rely on discrete signals (e.g., checking for text presence in the text rendering task), PickScore provides continuous rewards that capture both image details and aesthetic quality. This complexity demands denser optimization signals. TP-GRPO meets this requirement by capturing short- and long-term effects, leading to superior improvements over Flow-GRPO.
>
> [Q3] For the human preference task, images were selected randomly. For the two easier tasks, we first excluded samples where both Flow-GRPO and our method reached the maximum score of 1.00, as these cases do not highlight differences. For the remaining prompts, we then used random sampling. The table shows the ratio of samples achieving a score of 1.00 on the text rendering task:
>
> | Method                   | Ratio |
> | ------------------------ | ----- |
> | TP-GRPO (w/o constraint) | 0.665 |
> | TP-GRPO (w constraint)   | 0.674 |
>
> The high ratio confirms that "easy" samples dominate the test set, which masks potential gains and supports our first part in W2. For more qualitative comparison, please refer to Appendix F.
>
> [Q4] Please see our analysis in W3.
>
> ---
>
> [1] Liu, Jie, et al. "Flow-grpo: Training flow matching models via online rl." arXiv:2505.05470.
>
> [2] He, Xiaoxuan, et al. "Tempflow-grpo: When timing matters for grpo in flow models." arXiv:2508.04324.

---

> > ### Author Rebuttal · Reviewer_wofg · 2026-04-03
> >
> > I thank the authors for their thorough response. I appreciate the additional details regarding computational cost and the performance with respect to wall clock time. I believe adding them to the paper will strengthen the contribution. The additional explanation of W3 as enabling time-injective reward assignment is helpful.
> >
> > A couple of points:
> > - It would be beneficial to test in harder tasks if some of the benchmarks being used are essentially saturated.
> > - I generally buy the notion of turning points as a crude mechanism for more general credit assignment though it feels like this could be tested more thoroughly and alternative designs could be considered.
> >
> > I have raised my score accordingly.

---

> > > ### Author Response · Authors · 2026-04-03
> > >
> > > We sincerely thank you for your valuable suggestions and supportive feedback. Your previous comments have significantly contributed to improving the quality of our manuscript, and we will incorporate them into the final version.
> > >
> > > Regarding your two points:
> > >
> > > - More challenging tasks: Our current experiments are based on Flow-GRPO, which is widely adopted in the community. We agree that further validation is beneficial, and will conduct experiments on more challenging tasks further.
> > > - About the turning point mechanism: We appreciate your insight. We view this paper as a foundation for understanding turning points in diffusion-based RL and plan to investigate other influential factors, e.g., noise scale, to refine the framework in future work. We will also include a discussion on the papers you recommended in our revision.
> > >
> > > We believe the final version will fully address your requirements. Should you have any additional suggestions, we are more than happy to engage in further discussions and make any necessary improvements to the paper. :)
> > >
> > > Best regards,
> > >
> > > Authors

---

### Decision · Program_Chairs · 2026-04-30

**Decision:**

Accept (regular)

**Comment:**

This paper introduces TurningPoint GRPO (TP GRPO) to address reward sparsity and trajectory dependencies in Flow Matching alignment. The core contributions are a subtraction based stepwise reward and a novel turning point mechanism that captures long term credit assignment by identifying pivotal denoising steps.
While reviewers initially questioned the computational overhead and novelty relative to contemporaneous work, the rebuttal successfully demonstrated that TP GRPO achieves superior performance under equal wall clock time. Technical concerns regarding noise level bias were resolved by clarifying the ODE SDE branching strategy. Strong empirical gains on complex human preference tasks confirm the effectiveness of the method. The paper is technically sound and provides a distinct contribution to trajectory level modeling in diffusion based RL.
The paper received final review scores of 5, 4, and 5. The AC agrees with the consensus regarding the technical merit and empirical strength of the work and recommends acceptance.